# Learning Stochastic Bridges for Video Object Removal via Video-to-Video Translation

Zijie Lou [* 1]   Xiangwei Feng [* 1]   Jiaxin Wang [1 2]   Jiangtao Yao [1]   Fei Che [1]   Tianbao Liu [1]   Chengjing Wu [1]
Xiaochao Qu [1]   Luoqi Liu [1]   Ting Liu [1]

## Abstract

Existing video object removal methods predominantly rely on diffusion models following a noise-to-data paradigm, where generation starts from uninformative Gaussian noise. This approach discards the rich structural and contextual priors present in the original input video. Consequently, such methods often lack sufficient guidance, leading to incomplete object erasure or the synthesis of implausible content that conflicts with the scene's physical logic. In this paper, we reformulate video object removal as a video-to-video translation task via a stochastic bridge model. Unlike noise-initialized methods, our framework establishes a direct stochastic path from the source video (with objects) to the target video (objects removed). This bridge formulation effectively leverages the input video as a strong structural prior, guiding the model to perform precise removal while ensuring that the filled regions are logically consistent with the surrounding environment. To address the trade-off where strong bridge priors hinder the removal of large objects, we propose a novel adaptive mask modulation strategy. This mechanism dynamically modulates input embeddings based on mask characteristics, balancing background fidelity with generative flexibility. Extensive experiments demonstrate that our approach significantly outperforms existing methods in both visual quality and temporal consistency. The project page is https://bridgeremoval.github.io/.

## 1. Introduction

Video object removal (Kim et al., 2019; Li et al., 2020; Zou et al., 2021; Zhang et al., 2022; Zhou et al., 2023b; Ju et al., 2024; Li et al., 2025b) aims to erase specific objects from a video sequence while plausibly synthesizing the missing regions based on the surrounding context. The core challenge lies in reconstructing occluded areas in a manner that ensures spatial coherence within individual frames and temporal consistency across the entire sequence.

Prior works on this task have generally followed two primary directions: propagation-based methods (Kim et al., 2019; Li et al., 2020; Zou et al., 2021; Zhou et al., 2023b; Zhang et al., 2022) and generative inpainting methods (Ju et al., 2024; Li et al., 2025b; Miao et al., 2025). Propagation-based techniques typically rely on optical flow to transfer pixels from non-occluded frames to fill the missing regions. While effective for static scenes, these methods often struggle in complex scenarios involving dynamic camera movements or prolonged occlusions where valid source pixels are unavailable. Conversely, generative approaches formulate object removal as a synthesis problem. The recent emergence of large-scale diffusion models (Podell et al., 2023; Esser et al., 2024; Labs, 2024; Li et al., 2025b) has significantly advanced this domain, enabling the generation of high-fidelity textures and complex structures that were previously unattainable with traditional methods.

Despite these advancements, state-of-the-art video inpainting methods (Li et al., 2025b; Jiang et al., 2025; Miao et al., 2025), particularly those based on diffusion models, predominantly operate under a noise-to-data paradigm. In this standard framework, the generation process initiates from uninformative Gaussian noise and is tasked with denoising this stochastic state into a coherent video sequence conditioned on the masked input. We argue that this formulation is suboptimal for object removal task. Initiating generation from pure noise requires the model to synthesize the entire video content from scratch, often leading to leading to incomplete object erasure or the synthesis of implausible content that conflicts with the scene's physical logic To address these stability issues, a common strategy involves reference-guided generation, where a specialized image in-

---

[*]Equal contribution [1]MT Lab, Meitu Inc., Beijing 100083, China [2]Beijing Jiaotong University, Beijing 100044, China. Correspondence to: Ting Liu <lt@meitu.com>.

*Proceedings of the $43^{rd}$ International Conference on Machine Learning*, Seoul, South Korea. PMLR 306, 2026. Copyright 2026 by the author(s).

painting model processes a keyframe (usually the first frame) that subsequently serves as a guidance signal (Ouyang et al., 2024; Gao et al., 2025). However, this approach introduces significant drawbacks. It increases system complexity by necessitating an auxiliary image edit model and creates a dependency bottleneck where artifacts in the initial frame propagate through the sequence, potentially conflicting with the natural temporal evolution of the background.

In this work, inspired by recently proposed bridge models (Chen et al., 2021; Liu et al., 2023; Chen et al., 2023; Wang et al., 2025), we propose a paradigm shift by reformulating video object removal as a video-to-video translation task. We posit that the input video, even containing the unwanted object, holds a vast amount of valid structural and environmental information that should be leveraged as a strong prior rather than discarded in favor of Gaussian noise. To realize this, we introduce BridgeRemoval, a novel framework based on the stochastic bridge formulation (Tong et al., 2024; Zhou et al., 2023a; Chen et al., 2023) . Unlike standard diffusion-based methods that traverse a path from noise to data, our approach constructs a direct stochastic bridge between the source video distribution (the input video containing the object) and the target video distribution (the clean video with the object removed). By compressing videos into continuous latent representations via a Variational Autoencoder (VAE), we establish a tractable SDE-based generation process where the input latent representation serves directly as the prior boundary distribution.

This bridge formulation anchors the generative trajectory to the source video, naturally enforcing strong spatio-temporal coherence. The model learns a transformation that selectively modifies the object region while preserving the integrity of the unmasked context. Nevertheless, a direct application of this strong prior presents a trade-off. While excellent for background preservation, an overly rigid adherence to the source video can hinder the removal of large objects, as the model may struggle to deviate sufficiently from the input to synthesize large missing regions. To mitigate this, we propose a novel adaptive mask modulation (AMM) strategy. This mechanism dynamically adjusts the influence of the input embeddings based on the spatial characteristics of the mask, allowing the model to relax structural constraints in regions with large occlusions while maintaining strict fidelity in regions with valid backgrounds.

In summary, our contributions are as follows:

- We propose BridgeRemoval, a novel video-to-video generative framework that utilizes an SDE-based bridge model to reformulate video object removal. By establishing a direct stochastic path from the source video to the target, our method effectively leverages input priors to achieve better object removal.

- We introduce an adaptive mask modulation strategy that dynamically balances the trade-off between background fidelity and generative flexibility, ensuring robust performance across objects of varying scales.

- To provide a comprehensive evaluation of video object removal models, we propose BridgeRemoval-Bench, which encompasses a wide variety of scenarios and provides meticulously annotated masks.

- Extensive experiments demonstrate the effectiveness of our approach in diverse real-world scenarios, outperforming existing methods in both visual quality and temporal coherence, while also highlighting the potential of our framework to be extended to other video editing tasks.

## 2. Related Works

**Diffusion-based Video Object Removal** Diffusion-based methods (Li et al., 2025b; Zi et al., 2025a; Miao et al., 2025; Jiang et al., 2025; Lee et al., 2025) have achieved state-of-the-art performance in video object removal. DiffuEraser (Li et al., 2025b) adapts the image inpainting architecture of BrushNet (Ju et al., 2024) to the video domain via a progressive two-stage training strategy involving prior propagation and temporal refinement. Beyond simple occlusion handling, ROSE (Miao et al., 2025) addresses complex environmental interactions by leveraging a synthetic dataset with comprehensive supervision to simultaneously remove objects and their collateral visual effects, such as shadows and reflections. Furthermore, unified editing frameworks have also shown promise in this area. VACE (Jiang et al., 2025) proposes a universal backbone for diverse visual conditions, while GenOmnimatte (Lee et al., 2025) employs layered video decomposition, both of which support object removal as a downstream application.

**Bridge Models** Recently, bridge models (Chen et al., 2021; Tong et al., 2024; Liu et al., 2023; Zhou et al., 2023a; Chen et al., 2023; Zheng et al., 2025; He et al., 2024; De Bortoli et al., 2021; Peluchetti, 2023) have garnered significant attention for their ability to transcend the limitations of the Gaussian prior inherent in standard diffusion models. By establishing a direct stochastic path between two arbitrary distributions, bridge models have demonstrated the clear superiority of a data-to-data generation paradigm over traditional noise-to-data approaches, particularly in tasks such as image-to-image translation (Liu et al., 2023; Zhou et al., 2023a) and speech synthesis (Chen et al., 2023; Li et al., 2025a). Notably, FrameBridge (Wang et al., 2025) successfully extended this methodology to image-to-video (I2V) synthesis by introducing SNR-aligned fine-tuning to stabilize the learning process.

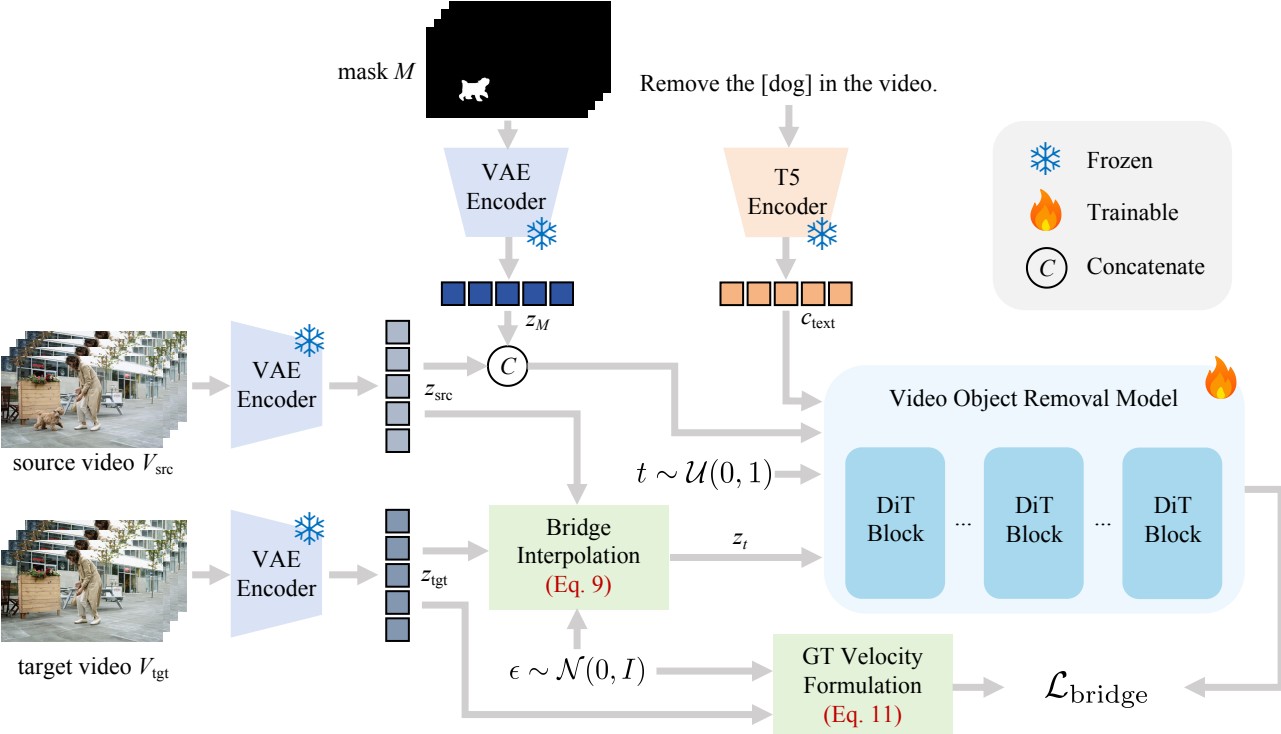

*Figure 1.* The framework of BridgeRemoval. The video inputs are projected into latent space using a frozen VAE encoder. Unlike standard diffusion, we employ a VP-SDE Bridge formulation to interpolate a trajectory ($z_t$) directly from the source video prior ($z_{\text{src}}$) to the clean target ($z_{\text{tgt}}$). The DiT-based model is conditioned on text embeddings ($c_{\text{text}}$) and a spatial input formed by concatenating the mask ($z_{\text{M}}$) with the source latent ($z_{\text{src}}$), optimized via a velocity-matching objective ($\mathcal{L}_{\text{bridge}}$).

## 3. Preliminaries

Probability path modeling (De Bortoli et al., 2021; Lipman et al., 2022; Liu et al., 2022) defines a class of generative models that describe continuous-time processes transporting mass from a prior distribution $p_1$ (source) to a target distribution $p_0$ (target). Generally, these models are governed by a stochastic differential equation (SDE) (Song et al., 2021):

$$dX_t = v(X_t, t)\,dt + g(t)\,dW_t, \quad t \in [0, 1], \quad (1)$$

where $v(X_t, t)$ is the velocity field driving the deterministic trajectory, $g(t)$ controls the stochastic diffusion, and $W_t$ is a standard Brownian motion.

### 3.1. Brownian Bridge

The standard Brownian Bridge (Albergo et al., 2025; Li et al., 2023) serves as a foundational instance of such processes, conditioned to start at a specific point $x_1$ and end at $x_0$. Unlike standard diffusion which maps data to Gaussian noise, a Brownian Bridge interpolates between two data points. Assuming a constant diffusion coefficient, the intermediate state $X_t$ follows a simple linear interpolation perturbed by noise:

$$X_t = (1-t)x_0 + tx_1 + \sqrt{t(1-t)}\epsilon, \quad \epsilon \sim \mathcal{N}(0, I). \quad (2)$$

While intuitive, this symmetric variance schedule $t(1-t)$ is rigid. It lacks the flexibility to handle complex high-dimensional data like video, where the signal-to-noise ratio (SNR) needs to be carefully scheduled to ensure stable training and generation.

### 3.2. VP-SDE Bridge

To overcome the limitations of the standard Brownian Bridge, we adopt a Variance-Preserving (VP) formulation inspired by the VP-SDE typically used in score-based generative models (Song et al., 2021). This formulation generalizes the bridge process by introducing time-dependent coefficients derived from a flexible noise schedule $\beta(t) \in [\beta_{\min}, \beta_{\max}]$.

Unlike the heuristic linear interpolation, the VP-SDE Bridge is theoretically grounded. As derived in Appendix B.1, the marginal distribution of this process remains Gaussian, allowing us to sample the intermediate state $X_t$ directly via a reparameterized ternary interpolation:

$$X_t = a_t x_0 + b_t x_1 + c_t \epsilon, \quad \epsilon \sim \mathcal{N}(0, I). \quad (3)$$

Crucially, the coefficients $a_t, b_t, c_t$ are not arbitrary but are strictly determined by the cumulative variance schedule $\sigma_t$

of the VP-SDE:

$$a_t = \frac{\bar{\sigma}_t^2}{\sigma_1^2}, \quad b_t = \frac{\sigma_t^2}{\sigma_1^2}, \quad c_t = \frac{\bar{\sigma}_t \sigma_t}{\sigma_1}, \quad (4)$$

where $\sigma_1^2$ is the total variance at $t = 1$, and $\bar{\sigma}_t^2 = \sigma_1^2 - \sigma_t^2$. This rigorous formulation guarantees that boundary conditions $X_0 = x_0$ and $X_1 = x_1$ are satisfied, while providing a tunable trajectory well-suited for high-fidelity video generation.

# 4. BridgeRemoval

As shown in Figure 1, we propose BridgeRemoval, a novel framework for video object removal. Unlike standard diffusion models that generate content from a generic Gaussian prior, our method leverages the VP-SDE Bridge formulation to construct a direct probability path from the source video (with objects) to the target video (object removed). This approach ensures higher fidelity and better preservation of non-masked regions.

## 4.1. Latent Encoding and Dual-Role Conditioning

Given a training triplet consisting of a source video $V_\text{src}$, a target video $V_\text{tgt}$, and a binary mask $M$, we first project the visual data into a compressed latent space using a pre-trained VAE encoder $\mathcal{E}$ (Wan et al., 2025):

$$z_\text{src} = \mathcal{E}(V_\text{src}), \quad z_\text{tgt} = \mathcal{E}(V_\text{tgt}), \quad z_M = \mathcal{E}(M). \quad (5)$$

Simultaneously, the text prompt $P$ is encoded into embeddings $c_\text{text}$ via a T5 encoder (Raffel et al., 2020).

To provide precise spatial guidance for inpainting, we construct a conditional input $y$ by concatenating the encoded mask with the source latent:

$$y = \text{Concat}(z_M, z_\text{src}). \quad (6)$$

It is crucial to note that $z_\text{src}$ plays a dual role in our framework. First, it acts as the *prior distribution* (the starting point of the bridge at $t = 1$). Second, it serves as a *spatial condition* in $y$, providing the network with contextual information about the background that should be preserved. This dual utilization effectively anchors the generation process, ensuring seamless blending between the inpainted region and the original background.

## 4.2. Velocity-Driven Bridge Training

To learn the transition trajectory between the source and target distributions, we frame the video removal task as a stochastic process governed by a Variance-Preserving Stochastic Differential Equation (VP-SDE) (Song et al., 2021).

**Bridge Process.** Standard diffusion models typically assume an uninformative Gaussian prior $\mathcal{N}(0, I)$. However, our task provides the source video $z_\text{src}$ as a strong structural prior. To leverage this, we adopt the Denoising Diffusion Bridge Model (DDBM) (Zhou et al., 2023a), essentially replacing the Gaussian prior with a Dirac prior $\delta_{z_\text{src}}$.

Specifically, the bridge process is defined by a forward SDE that drifts towards the source video $z_\text{src}$:

$$\begin{aligned} dz_t = &\left[f(t)z_t + g(t)^2 h(z_t, t, z_\text{src})\right] dt \\ &+ g(t)dw, \quad z_0 \sim p_\text{tgt}, \ z_T = z_\text{src}, \end{aligned} \quad (7)$$

where $f(t)$ and $g(t)$ are the drift and diffusion coefficients of the VP-SDE. The term $h(z_t, t, z_\text{src}) \triangleq \nabla_{z_t} \log p(z_\text{src}|z_t)$ acts as a guidance force, pulling the trajectory towards $z_\text{src}$.

Correspondingly, the generative reverse process is governed by:

$$\begin{aligned} dz_t = &[f(t)z_t - g(t)^2(s_\theta(z_t, t, z_\text{src}) \\ &- h(z_t, t, z_\text{src}))]dt + g(t)d\bar{w}, \end{aligned} \quad (8)$$

where $s_\theta(z_t, t, z_\text{src})$ approximates the score function. Since the transition kernel remains Gaussian, the term $h$ is analytically tractable. We provide the proof of the Gaussian nature of the bridge marginal (see Appendix B.1) and the derivation of $h$ (see Appendix B.2).

**Training Objective.** Instead of simulating stochastic paths directly, we leverage the analytic property of the bridge marginal distribution. As derived in Appendix B.1, the intermediate state $z_t$ can be sampled directly via a weighted ternary interpolation:

$$z_t = a_t z_\text{tgt} + b_t z_\text{src} + c_t \epsilon, \quad \epsilon \sim \mathcal{N}(0, I). \quad (9)$$

Here, $z_t$ is a mixture of the clean target $z_\text{tgt}$, the source prior $z_\text{src}$, and noise $\epsilon$. The scalar coefficients $a_t, b_t, c_t$ are rigorously derived from the VP-SDE variance schedule $\sigma_t$ (see Appendix B.1):

$$a_t = \frac{\bar{\sigma}_t^2}{\sigma_1^2}, \quad b_t = \frac{\sigma_t^2}{\sigma_1^2}, \quad c_t = \frac{\bar{\sigma}_t \sigma_t}{\sigma_1}. \quad (10)$$

These coefficients ensure the boundary conditions $z_0 = z_\text{tgt}$ and $z_1 = z_\text{src}$.

To train the network $v_\theta$ parameterized by $\theta$, we define a ground-truth velocity target $u_t$. As shown in Appendix B.3, minimizing the velocity matching error is equivalent to score matching:

$$u_t = \frac{a_t}{\rho_t}\epsilon - \frac{c_t}{\rho_t}z_\text{tgt}, \quad \text{where } \rho_t = \sqrt{a_t^2 + c_t^2}. \quad (11)$$

The network is optimized by minimizing the mean squared error:

$$\mathcal{L}_\text{bridge} = \mathbb{E}_{t, \epsilon, z_\text{src}, z_\text{tgt}}\left[\|v_\theta(z_t, t, y, c_\text{text}) - u_t\|^2\right]. \quad (12)$$

The complete training procedure is detailed in Algorithm 1.

**Algorithm 1** Training

**Input:** source video $V_{src}$, target video $V_{tgt}$, mask $M$, text prompt $P$, VAE encoder $\mathcal{E}$, T5 encoder, model $v_\theta$, variance schedule $\sigma_t$

1 **repeat**
2     Encode latents $z_{src} \leftarrow \mathcal{E}(V_{src})$, $z_{tgt} \leftarrow \mathcal{E}(V_{tgt})$, and $z_M \leftarrow \mathcal{E}(M)$
3     Encode text embeddings $c_{text} \leftarrow$ T5$(P)$
4     Construct conditional input $y \leftarrow$ Concat$(z_M, z_{src})$
5     Sample interpolation time $t \sim \mathcal{U}(0,1)$ and noise $\epsilon \sim \mathcal{N}(0, I)$
6     Compute SDE coefficients $a_t, b_t, c_t$ based on the variance schedule $\sigma_t$
7     Construct intermediate bridge state $z_t = a_t z_{tgt} + b_t z_{src} + c_t \epsilon$
8     Compute normalization factor $\rho_t = \sqrt{a_t^2 + c_t^2}$
9     Compute velocity target $u_t = \frac{a_t}{\rho_t}\epsilon - \frac{c_t}{\rho_t}z_{tgt}$
10     Update parameters $\theta$ by gradient descent on $\|v_\theta(z_t, t, y, c_{text}) - u_t\|^2$
11 **until** *convergence*;

---

**Algorithm 2** Inference

**Input:** source video $V_{src}$, mask $M$, text prompt $P$, trained model $v_\theta$, steps $N$, variance schedule $\sigma_t$, T5 encoder, VAE encoder $\mathcal{E}$, VAE decoder $\mathcal{D}$

1 Encode input latents $z_{src} \leftarrow \mathcal{E}(V_{src})$ and condition $y \leftarrow$ Concat$(\mathcal{E}(M), z_{src})$
2 Encode text embeddings $c_{text} \leftarrow$ T5$(P)$
3 Initialize state $z \leftarrow z_{src}$ using the source video as the prior
4 Define time schedule $\{t_k\}_{k=N}^0$ from 1 to 0 **for** $k = N, \ldots, 1$ **do**
5     Set current time $t \leftarrow t_k$ and next time $t' \leftarrow t_{k-1}$
6     Predict velocity field $\hat{v}_t \leftarrow v_\theta(z, t, y, c_{text})$
7     Recover predicted target $\hat{z}_{0|t}$ via Eq. 13
8     **if** $k > 1$ **then**
9        Compute SDE solver weights $w_1, w_2, w_3$ based on $\sigma_t$
10        Sample noise $\epsilon' \sim \mathcal{N}(0, I)$
11        Update state $z \leftarrow w_1 z + w_2 \hat{z}_{0|t} + w_3 \epsilon'$
12     **else**
13        Set final state $z \leftarrow \hat{z}_{0|t}$
14     **end**
15 **end**
16 Decode output video $V_{out} \leftarrow \mathcal{D}(z)$
**Output:** Inpainted video $V_{out}$

## 4.3. Generative Inference via SDE Solver

During inference, we aim to reverse the bridge process to traverse from the source latent $z_{src}$ back to the clean target $z_{tgt}$. Theoretically, this corresponds to solving the backward SDE (Eq. 8) derived in Sec. 4.2. Since the continuous backward SDE involves the unknown score function $s_\theta$, we employ a numerical SDE solver to approximate the trajectory. We adopt a variance-corrected sampling strategy that acts as a discretization of the backward SDE, ensuring the generation strictly follows the statistics of the Brownian bridge.

Given the current state $z_t$ at timestep $t$, the network predicts the velocity field $\hat{v}_t = v_\theta(z_t, t, y)$, which approximates the drift term of the reverse process. We first estimate the clean target latent $\hat{z}_{0|t}$ by inverting the velocity equation (Eq. 11), utilizing the normalization factor $\rho_t = \sqrt{a_t^2 + c_t^2}$ defined previously:

$$\hat{z}_{0|t} = \frac{1}{\rho_t^2/c_t} \left( \frac{a_t}{c_t}z_t - \frac{a_t b_t}{c_t}z_{src} - \rho_t \hat{v}_t \right). \quad (13)$$

This formula explicitly removes the predicted noise and the weighted source component from $z_t$, recovering an approximation of the target $z_{tgt}$. To progress to the next timestep $t' < t$, we perform an SDE solver step that combines the current state $z_t$, the predicted target $\hat{z}_{0|t}$, and Gaussian noise injection:

$$z_{t'} = w_1 z_t + w_2 \hat{z}_{0|t} + w_3 \epsilon', \quad \epsilon' \sim \mathcal{N}(0, I). \quad (14)$$

Here, $w_1, w_2, w_3$ are scalar weights analytically determined by the bridge schedule to ensure the trajectory maintains the correct conditional variance. Specifically, $w_3$ modulates

the contribution of the freshly sampled noise $\epsilon'$ to prevent variance collapse near the target. The detailed inference algorithm is summarized in Algorithm 2.

## 4.4. Adaptive Mask Modulation

While the bridge formulation effectively anchors the generation trajectory to the source video, this rigid adherence to the source distribution can become detrimental when removing large objects. To mitigate this limitation, we propose Adaptive Mask Modulation (AMM), a spatially-aware feature modulation mechanism inspired by FiLM (Perez et al., 2018). The mask $M \in \mathbb{R}^{C_m \times F \times H \times W}$ is then projected into a latent feature space through a learnable patch embedding:

$$F_M = \phi_M(M) \in \mathbb{R}^{D \times F' \times H' \times W'} \quad (15)$$

where $\phi_M$ is a 3D convolutional layer. From the mask features $F_M$, we predict spatially-varying modulation parameters through two $1 \times 1 \times 1$ convolutional projections:

$$\gamma = f_\gamma(F_M), \quad \beta = f_\beta(F_M) \quad (16)$$

where $\gamma, \beta \in \mathbb{R}^{D \times F' \times H' \times W'}$ represent the scale and shift parameters, respectively.

Let $h \in \mathbb{R}^{D \times F' \times H' \times W'}$ denote the input patch embeddings after the initial projection. The modulated embeddings $h'$ are computed as:

$$h' = h \odot (1 + \gamma) + \beta \quad (17)$$

Scale modulation $(1 + \gamma)$ adaptively amplifies or attenuates features based on their spatial relationship to the mask, enabling the model to learn where to preserve original information. Shift modulation $(\beta)$ injects position-aware guidance into the feature, providing explicit spatial cues about the inpainting region boundaries and structure.

By incorporating rich spatial mask information through learnable modulation, AMM enables the model to adaptively adjust its behavior across different spatial regions, effectively counterbalancing the strong source prior within masked regions while maintaining strict fidelity to the surrounding environment.

## 5. Experiments

### 5.1. Experimental Setup

**Training Data.** We construct a training dataset totaling approximately 47,000 video pairs from two complementary sources. First, we utilize the ROSE dataset (Miao et al., 2025), which consists of 16,000 synthetically generated samples. By leveraging 3D engines to toggle object visibility, ROSE provides perfectly aligned triplets of source videos, target backgrounds, and masks. While this offers high-quality supervision, our statistical analysis indicates a distribution bias towards static, rigid objects, limiting the model's generalization to complex motion. To address this limitation, we introduce a supplementary real-world composite dataset comprising 31,000 video pairs. This subset is explicitly designed to improve robustness on dynamic agents, containing 15,000 human-centric videos and 16,000 videos of common dynamic objects (e.g., animals, vehicles). The data pipeline adopts a composition-based approach: we employ SAM2 (Ravi et al., 2025) to extract high-fidelity foregrounds from real-world videos and composite them onto clean backgrounds. A Qwen3-VL (Bai et al., 2025b) based filtering mechanism is then applied to ensure visual coherence and discard low-quality samples.

**Testing Data.** Existing video object removal benchmarks suffer from limited scale and insufficient scene diversity. For instance, DAVIS (Perazzi et al., 2016) contains only 50 videos, all in landscape orientation, with some challenging cases involving fast motion. ROSE-Bench (Miao et al., 2025) includes 60 videos, but most depict static scenes, and its publicly released dataset does not specify a test split, making it unsuitable for standardized evaluation.

To enable comprehensive assessment of video object removal models, we introduce BridgeRemoval-Bench, a new benchmark comprising 150 high-quality videos with rich scene variety, including both landscape and portrait orientations, as well as annotations for both single and multiple objects. The dataset is constructed from three distinct sources:

- 50 real-world videos collected from stock platforms such as Pexels (pex), featuring aesthetically pleasing visuals and clean backgrounds.

- 50 synthetic videos generated by first using Qwen3 (Yang et al., 2025a) to produce diverse text prompts and then synthesizing videos with Veo (veo), covering common categories including humans, animals, and vehicles.

- 50 in-the-wild videos sourced from social media platforms like TikTok (tik), exhibiting highly complex and challenging scenarios.

For all 150 videos, object masks are generated using SAM2 (Ravi et al., 2025). Furthermore, to support methods such as VACE (Jiang et al., 2025) that require a textual description of the target video (object removed), we employ Qwen3-VL (Bai et al., 2025b) to generate a detailed caption for each video (object removed), and performed the same process on the DAVIS dataset. In summary, the DAVIS dataset and BridgeRemoval-Bench are used to evaluate the performance of various algorithms.

**Metrics.** To comprehensively evaluate the performance of each model, we employ the following metrics:

- CLIP-T (Bai et al., 2025a) is used to calculate the similarity between the result video and the text description to verify whether the object has been removed. And CLIP-F (Bai et al., 2025a) calculates the average inter-frame CLIP similarity to gauge temporal consistency.

- Following prior works (Bian et al., 2025), we evaluate the fidelity of the background using standard metrics, including PSNR, MSE, and LPIPS (Zhang et al., 2018). These metrics are computed exclusively on the unmasked areas.

- We also report results from VBench (Huang et al., 2024). However, as these metrics from VBench exhibit limited discriminative power in our specific task and may not fully reflect the nuanced performance of the algorithms, we provide these results in the Appendix D for reference only.

**Implementation Details.** Our model is initialized from the pre-trained Wan2.1-1.3B (Wan et al., 2025). During training, we utilize video clips consisting of 81 frames. To accommodate varying aspect ratios, we employ a bucket-based multi-resolution strategy, where videos are resized to resolutions such as $576 \times 1024$, $1024 \times 576$, or $768 \times 768$ based on their original orientation. The bridge process is parameterized with 1000 discrete timesteps during training, utilizing a linear variance schedule with $\beta_{\min} = 0.01$ and $\beta_{\max} = 50.0$. We employ the AdamW optimizer (Loshchilov & Hutter,

*Table 1.* Quantitative comparisons among BridgeRemoval and other video object removal models in DAVIS dataset. **Red** stands for the best, **Blue** stands for the second best.

| Method | Year-Venue | Base Model | Video Quality | | Unmasked Region Preservation | | | Runtime (s/frame) ↓ |
|---|---|---|---|---|---|---|---|---|
| | | | CLIP-T ↑ | CLIP-F ↑ | PSNR ↑ | MSE ↓ | LPIPS ↓ | |
| Senoria (Zi et al., 2025b) | 2025-arxiv | CogVideoX-5B (Yang et al., 2025b) | 0.2618 | 0.9654 | 17.4752 | 2057.1134 | 0.4654 | 2.143 |
| Ditto (Bai et al., 2025a) | 2025-arxiv | Wan2.1-14B (Wan et al., 2025) | 0.2196 | **0.9712** | 11.7304 | 4945.1432 | 0.5226 | 6.402 |
| ICVE (Liao et al., 2025) | 2025-arxiv | Hunyuanvideo-13B (Kong et al., 2024) | 0.2361 | 0.9670 | 25.9675 | 244.8118 | 0.1116 | 12.571 |
| ROSE (Miao et al., 2025) | 2025-NeurIPS | Wan2.1-1.3B (Wan et al., 2025) | 0.2353 | 0.9676 | 26.1202 | 204.0729 | 0.1046 | **0.741** |
| VACE (Jiang et al., 2025) | 2025-ICCV | Wan2.1-1.3B (Wan et al., 2025) | 0.2629 | 0.9654 | **27.3758** | **138.2393** | **0.0796** | 1.765 |
| GenOmnimatte (Lee et al., 2025) | 2025-CVPR | Wan2.1-1.3B (Wan et al., 2025) | **0.2814** | 0.9701 | 26.6812 | 177.3141 | 0.125 | 3.914 |
| BridgeRemoval (Ours) | 2026 | Wan2.1-1.3B (Wan et al., 2025) | **0.2788** | **0.9768** | **28.2154** | **133.3605** | **0.0977** | **1.111** |

*Table 2.* Quantitative comparisons among BridgeRemoval and other video object removal models in BridgeRemoval-Bench. **Red** stands for the best, **Blue** stands for the second best.

| Method | Year-Venue | Base Model | Video Quality | | Unmasked Region Preservation | | | Runtime (s/frame) ↓ |
|---|---|---|---|---|---|---|---|---|
| | | | CLIP-T ↑ | CLIP-F ↑ | PSNR ↑ | MSE ↓ | LPIPS ↓ | |
| Senoria (Zi et al., 2025b) | 2025-arxiv | CogVideoX-5B (Yang et al., 2025b) | 0.2920 | 0.9843 | 18.9123 | 1553.9859 | 0.3365 | 2.143 |
| Ditto (Bai et al., 2025a) | 2025-arxiv | Wan2.1-14B (Wan et al., 2025) | 0.2218 | 0.9873 | 10.5340 | 7523.6043 | 0.5262 | 6.402 |
| ICVE (Liao et al., 2025) | 2025-arxiv | Hunyuanvideo-13B (Kong et al., 2024) | 0.2842 | 0.9887 | 17.4098 | 1589.3646 | 0.2740 | 12.571 |
| ROSE (Miao et al., 2025) | 2025-NeurIPS | Wan2.1-1.3B (Wan et al., 2025) | 0.2219 | **0.9879** | 30.5175 | 84.6950 | 0.0470 | **0.741** |
| VACE (Jiang et al., 2025) | 2025-ICCV | Wan2.1-1.3B (Wan et al., 2025) | 0.2875 | 0.9859 | 31.0548 | **61.7623** | **0.0335** | 1.765 |
| GenOmnimatte (Lee et al., 2025) | 2025-CVPR | Wan2.1-1.3B (Wan et al., 2025) | **0.2966** | 0.9878 | **31.5999** | 63.3032 | 0.0506 | 3.914 |
| BridgeRemoval (Ours) | 2026 | Wan2.1-1.3B (Wan et al., 2025) | **0.2922** | **0.9917** | **33.2394** | **46.3525** | **0.0415** | **1.111** |

2019) with a constant learning rate of $2 \times 10^{-5}$ and train the model for 20000 steps on 8 NVIDIA H100 GPUs. For inference, we use a fast sampling strategy with 50 steps to ensure efficient generation.

## 5.2. Quantitative Evaluation

We compared BridgeRemoval with six state-of-the-art methods including Senoria (Zi et al., 2025b), Ditto (Bai et al., 2025a), ICVE (Liao et al., 2025), ROSE (Miao et al., 2025), VACE (Jiang et al., 2025), and GenOmnimatte (Lee et al., 2025) on both DAVIS (Perazzi et al., 2016) and BridgeRemoval-Bench. CLIP-T and CLIP-F measure the model's capability to remove the target object, while PSNR, MSE, and LPIPS evaluate its ability to preserve the background content. A high-performing video object removal method must excel in both aspects. Thanks to the stochastic bridge formulation, BridgeRemoval achieves strong performance in both object removal and background preservation.

As shown in Table 1 and 2, our method attains the best or second-best results across all quantitative metrics. Notably, on BridgeRemoval-Bench, our approach outperforms the current state-of-the-art method by 1.64 dB in PSNR. In contrast, other methods exhibit a clear trade-off. Some successfully remove the object but fail to preserve the background, while others maintain the background well but inadequately remove the target. For instance, VACE (Jiang et al., 2025) tends to generate a new object within the masked region rather than seamlessly inpainting the background. Conse-

quently, it achieves relatively high scores in PSNR, MSE, and LPIPS but performs poorly in CLIP-T and CLIP-F, indicating ineffective object removal. In addition, we compared the inference speed of different methods. All methods are evaluated on L20X GPUs, our method runs at 1.111 seconds per frame, striking a favorable balance between generation quality and inference efficiency.

## 5.3. Qualitative Comparison

For the visual comparison, we compare our method with ROSE (Miao et al., 2025), VACE (Jiang et al., 2025), and GenOmnimatte (Lee et al., 2025), which are representative methods of diffusion-based approaches. Figure 2 presents four comparison results for video object removal. Our method starts from the source video rather than random noise and leverages a strong prior to effectively fill the masked regions. It completes masked regions with coherence and clear contents, while other compared methods tend to fail or produce unpleasant inpainting results such as texture distortions and black hazy region in ROSE results, as well as artifacts in VACE and GenOmnimatte. More samples can be visited at: https://bridgeremoval.github.io/.

## 5.4. Ablation Study

We conduct comprehensive ablation studies to validate the design choices of our framework. Detailed results are provided in Appendix C.

masked frames ROSE VACE GenOmnimatte Ours

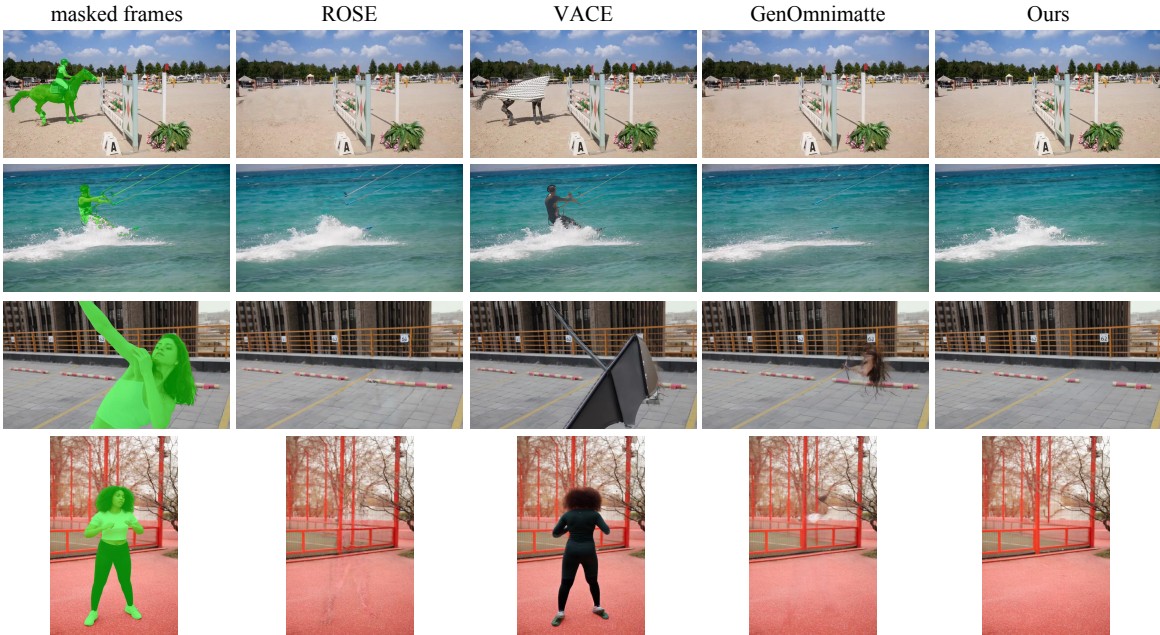

*Figure 2.* Qualitative comparisons. The input videos in the first two rows are from DAVIS dataset, and those in the last two rows are from BridgeRemoval-Bench.

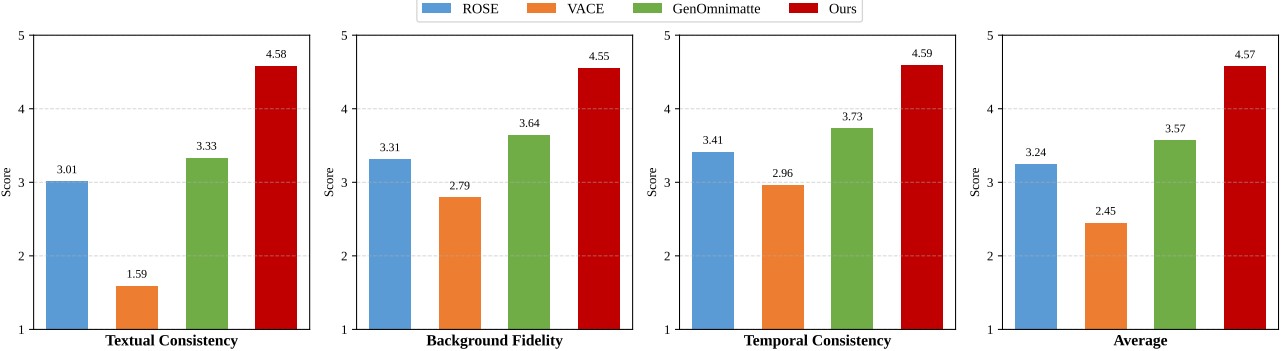

*Figure 3.* User study. Users rated BridgeRemoval and other video object removal methods based on three criteria: text consistency, background fidelity, and temporal consistency.

## 5.5. User Study

We conducted a user study involving 24 participants, including researchers and students specializing in image and video editing, to evaluate randomly selected cases. The assessment focuses on three dimensions: Textual Consistency (whether the target is effectively removed), Background Fidelity (how well the background is preserved), and Temporal Consistency (the presence of flickering or artifacts). Each dimension is rated on a 5-point scale (1: lowest, 5: highest). A total of 1440 valid evaluations were collected. As shown in Figure 3, BridgeRemoval significantly outperformed existing baselines, achieving higher preference rates across all evaluation criteria.

## 6. Conclusion

This study introduces BridgeRemoval, a stochastic bridge framework that reformulates video object removal as a video-to-video translation task. By leveraging structural priors from the input video and employing an adaptive mask modulation strategy, our method effectively balances background fidelity with generative flexibility. Extensive experiments demonstrate that our approach outperforms existing methods in visual quality and temporal consistency. This work validates the efficacy of bridge-based models for video object removal and we believe that the designs in BridgeRemoval will provide valuable insights to the video edit community.

## Impact Statement

This paper introduces BridgeRemoval, a Bridge-based framework advancing high-fidelity video object removal. While enhancing content creation and post-production, this technology poses potential risks for manipulating authentic footage or altering context. We emphasize the importance of responsible deployment and detection methods to mitigate the misuse of generative video editing tools.

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

## A. Pseudo Code for the Training and inference of BridgeRemoval

To clearly delineate the distinctions between BridgeRemoval and other models (such as ROSE (Miao et al., 2025), which is trained with the Flow Matching framework.) in video object removal, we present a detailed comparison of their algorithmic formulations. The detailed configurations are summarized in Table 3.

*Table 3.* Comparison of algorithmic settings between the Flow Matching Baseline and BridgeRemoval (Ours).

|  | **Flow Matching Baseline** | **BridgeRemoval (Ours)** |
|---|---|---|
| **Input Data** | source video $V_{\text{src}}$, target video $V_{\text{tgt}}$ (during training), mask $M$ | source video $V_{\text{src}}$, target video $V_{\text{tgt}}$ (during training), mask $M$ |
| **Condition** | $y = \text{Concat}(z_M, z_{\text{src}})$ | $y = \text{Concat}(z_M, z_{\text{src}})$ |
| **Training** | $z_t = (1-t)z_{\text{tgt}} + t\epsilon$ | $z_t = a_t z_{\text{tgt}} + b_t z_{\text{src}} + c_t \epsilon$ |
| **Prediction Target** | Velocity $v = \epsilon - z_{\text{tgt}}$ | Velocity $u_t = \frac{a_t}{\rho_t}\epsilon - \frac{c_t}{\rho_t}z_{\text{tgt}}$ |
| **Inference Init** | Gaussian Noise $\mathcal{N}(0, I)$ | Source Prior $z_{\text{src}}$ |
| **Trajectory** | ODE Integration ($z_1 \rightarrow z_0$) | Bridging ($z_{\text{src}} \rightarrow z_{\text{tgt}}$) |

We provide the pseudo code for the training and inference process of BridgeRemoval (See Algorithm 4 and 6). Meanwhile, we also provide that of Flow Matching-based video object removal models (See Algorithm 3 and 5) to show the distinctions between BridgeRemoval and Flow Matching-based models. In practical implementation, time-dependent coefficients (e.g., $\bar{\alpha}t$ or $a_t, b_t, c_t$) are typically precomputed and retrieved from a schedule table for computational efficiency.

---

**Algorithm 3** Flow Matching Training for Video Object Removal

**Input:** Source Video $V_{\text{src}}$, Target Video $V_{\text{tgt}}$, Mask $M$, Text Prompt $P$, Model $v_\theta$, VAE Encoder $\mathcal{E}$, T5 Encoder
**Output:** Optimized parameters $\theta$

1 Encode inputs to latent space: $z_{\text{src}} \leftarrow \mathcal{E}(V_{\text{src}})$, $z_0 \leftarrow \mathcal{E}(V_{\text{tgt}})$, and $z_M \leftarrow \mathcal{E}(M)$
2 Construct conditional input by concatenating mask and source video: $y \leftarrow \text{Concat}(z_M, z_{\text{src}})$
3 Encode text prompt to embeddings: $c_{\text{text}} \leftarrow \text{T5}(P)$
4 **repeat**
5      Sample continuous time $t \sim \mathcal{U}(0, 1)$ and Gaussian noise $\epsilon \sim \mathcal{N}(0, I)$
6      Construct interpolated state via linear interpolation: $z_t = (1-t)z_0 + t\epsilon$
7      Compute ground-truth velocity target: $v = \epsilon - z_0$
8      Predict velocity field using the model: $\hat{v} \leftarrow v_\theta(z_t, t, y, c_{\text{text}})$
9      Compute velocity matching loss: $\mathcal{L} = \|\hat{v} - v\|^2$
10      Update parameters $\theta$ by gradient descent
11 **until** *convergence*;

---

**Algorithm 4** BridgeRemoval Training for Video Object Removal

**Input:** Source Video $V_{\text{src}}$, Target Video $V_{\text{tgt}}$, Mask $M$, Text Prompt $P$, Model $v_\theta$, Variance Schedule $\sigma_t$, VAE Encoder $\mathcal{E}$, T5 Encoder
**Output:** Optimized parameters $\theta$

1 Encode inputs to latent space: $z_{\text{src}} \leftarrow \mathcal{E}(V_{\text{src}})$, $z_{\text{tgt}} \leftarrow \mathcal{E}(V_{\text{tgt}})$, and $z_M \leftarrow \mathcal{E}(M)$
2 Construct conditional input: $y \leftarrow \text{Concat}(z_M, z_{\text{src}})$
3 Encode text prompt to embeddings: $c_{\text{text}} \leftarrow \text{T5}(P)$
4 **repeat**
5      Sample interpolation time $t \sim \mathcal{U}(0, 1)$ and noise $\epsilon \sim \mathcal{N}(0, I)$
6      Compute SDE coefficients $a_t, b_t, c_t$ derived from the variance schedule $\sigma_t$
7      Construct intermediate bridge state via ternary interpolation: $z_t = a_t z_{\text{tgt}} + b_t z_{\text{src}} + c_t \epsilon$
8      Compute ground-truth velocity target $u_t$ (V-Prediction parameterization): $\rho_t = \sqrt{a_t^2 + c_t^2}$,    $u_t = \frac{a_t}{\rho_t}\epsilon - \frac{c_t}{\rho_t}z_{\text{tgt}}$
9      Predict velocity field using the model: $\hat{v}_t \leftarrow v_\theta(z_t, t, y, c_{\text{text}})$
10      Compute velocity matching loss: $\mathcal{L} = \|\hat{v}_t - u_t\|^2$
11      Update parameters $\theta$ by gradient descent
12 **until** *convergence*;

---

---

**Algorithm 5** Flow Matching Inference for Video Object Removal

---

**Input:** Source Video $V_{\text{src}}$, Mask $M$, Text Prompt $P$, Model $v_\theta$, Steps $N$, VAE Encoder $\mathcal{E}$, VAE Decoder $\mathcal{D}$

**Output:** Inpainted Video $V_{\text{out}}$

1 Encode inputs: $z_{\text{src}} \leftarrow \mathcal{E}(V_{\text{src}})$, $z_M \leftarrow \mathcal{E}(M)$ and condition $y \leftarrow \text{Concat}(z_M, z_{\text{src}})$
2 Encode text prompt: $c_{\text{text}} \leftarrow \text{T5}(P)$
3 Initialize latent state from pure Gaussian noise: $z \sim \mathcal{N}(0, I)$
4 Define time schedule $\{t_k\}_{k=N}^0$ from 1 to 0
5 **for** $k = N, \ldots, 1$ **do**
6     Set current time $t \leftarrow t_k$ and next time $t' \leftarrow t_{k-1}$
7     Predict velocity field: $\hat{v} \leftarrow v_\theta(z, t, y, c_{\text{text}})$
8     Compute step size: $\Delta t = t' - t$
9     Euler ODE integration step: $z \leftarrow z + \hat{v} \cdot \Delta t$
10 **end**
11 Decode output latent to pixel space: $V_{\text{out}} \leftarrow \mathcal{D}(z)$

---

**Algorithm 6** BridgeRemoval Inference for Video Object Removal

---

**Input:** Source Video $V_{\text{src}}$, Mask $M$, Text Prompt $P$, Model $v_\theta$, Steps $N$, Variance Schedule $\sigma_t$, VAE Encoder $\mathcal{E}$, VAE Decoder $\mathcal{D}$

**Output:** Inpainted Video $V_{\text{out}}$

1 Encode inputs: $z_{\text{src}} \leftarrow \mathcal{E}(V_{\text{src}})$, $z_M \leftarrow \mathcal{E}(M)$ and condition $y \leftarrow \text{Concat}(z_M, z_{\text{src}})$
2 Encode text prompt: $c_{\text{text}} \leftarrow \text{T5}(P)$
3 Initialize state $z \leftarrow z_{\text{src}}$ using the source video as the prior
4 Define time schedule $\{t_k\}_{k=N}^0$ from 1 to 0
5 **for** $k = N, \ldots, 1$ **do**
6     Set current time $t \leftarrow t_k$ and next time $t' \leftarrow t_{k-1}$
7     Predict velocity field: $\hat{v}_t \leftarrow v_\theta(z, t, y, c_{\text{text}})$
8     Recover predicted target $\hat{z}_{0|t}$ from velocity (analytic inversion): $\rho_t = \sqrt{a_t^2 + c_t^2}$, $\quad \hat{z}_{0|t} = \frac{a_t}{\rho_t^2} z - \frac{c_t}{\rho_t} \hat{v}_t - \frac{a_t b_t}{\rho_t^2} z_{\text{src}}$
9     **if** $k > 1$ **then**
10         Compute SDE solver weights $w_1, w_2, w_3$ based on $\sigma_t$
11         Sample noise $\epsilon' \sim \mathcal{N}(0, I)$
12         Update state via Euler solver: $z \leftarrow w_1 z + w_2 \hat{z}_{0|t} + w_3 \epsilon'$
13     **else**
14         Set final state directly: $z \leftarrow \hat{z}_{0|t}$
15     **end**
16 **end**
17 Decode output video: $V_{\text{out}} \leftarrow \mathcal{D}(z)$

---

## B. Mathematical Proofs

### B.1. Derivation of Bridge Marginal Distribution

To provide a theoretical foundation for the proposed BridgeRemoval and justify the coefficients used in Eq. 10, we first derive the marginal distribution of a general bridge process $p_{t,\text{bridge}}(\mathbf{z}_t | \mathbf{z}_0, \mathbf{z}_T)$ following the framework of DDBM (Zhou et al., 2023a).

In this section, we adopt standard diffusion notation (Ho et al., 2020), letting $\mathbf{z}_0$ represent the clean data (Target) and $\mathbf{z}_T$ represent the prior (Source). We explicitly use $\alpha_t, \sigma_t$ to denote the noise schedule coefficients of the underlying process. We will later map these general theoretical results to the specific $\sigma_t$-based parameterization used in our main text.

Using Bayes' rule and the Markov property of diffusion (Kingma et al., 2021) :

$$p_{t,\text{bridge}}(\mathbf{z}_t | \mathbf{z}_0, \mathbf{z}_T) = \frac{p(\mathbf{z}_T | \mathbf{z}_t) p(\mathbf{z}_t | \mathbf{z}_0)}{p(\mathbf{z}_T | \mathbf{z}_0)}. \tag{18}$$

Assuming the underlying process is a VP-SDE with Gaussian transition kernels:

$$p(\mathbf{z}_t|\mathbf{z}_0) = \mathcal{N}(\mathbf{z}_t; \alpha_t \mathbf{z}_0, \sigma_t^2 \mathbf{I}),$$

$$p(\mathbf{z}_T|\mathbf{z}_t) = \mathcal{N}\left(\mathbf{z}_T; \frac{\alpha_T}{\alpha_t}\mathbf{z}_t, \left(\sigma_T^2 - \frac{\alpha_T^2}{\alpha_t^2}\sigma_t^2\right)\mathbf{I}\right). \tag{19}$$

where $\text{SNR}_t = \alpha_t^2/\sigma_t^2$ (Kingma et al., 2021).

The resulting bridge marginal is also Gaussian (Zhou et al., 2023a) :

$$p_{t,\text{bridge}}(\mathbf{z}_t|\mathbf{z}_0, \mathbf{z}_T) = \mathcal{N}(\mathbf{z}_t; \boldsymbol{\mu}_t, \Sigma_t \mathbf{I}), \tag{20}$$

where the reparameterized state can be written as:

$$\mathbf{z}_t = A_t \mathbf{z}_0 + B_t \mathbf{z}_T + C_t \boldsymbol{\epsilon}, \quad \boldsymbol{\epsilon} \sim \mathcal{N}(0, \mathbf{I}), \tag{21}$$

with general coefficients:

$$A_t = \alpha_t \left(1 - \frac{\text{SNR}_T}{\text{SNR}_t}\right), \quad B_t = \frac{\text{SNR}_T}{\text{SNR}_t}\frac{\alpha_t}{\alpha_T}, \quad C_t = \sigma_t \sqrt{1 - \frac{\text{SNR}_T}{\text{SNR}_t}}. \tag{22}$$

Here we prove that the coefficients $a_t, b_t, c_t$ defined in Eq. 10 are a valid instance of the general theory derived above.

In the main text, we define a schedule variable $\sigma_t^2$ (let's denote it $\hat{\sigma}_t^2$ here to distinguish from diffusion variance) representing the cumulative variance contribution from the source. Specifically, we set:

$$b_t = \frac{\hat{\sigma}_t^2}{\hat{\sigma}_1^2}$$

Comparing this to the general form of $B_t$ in Eq. 22:

$$\frac{\hat{\sigma}_t^2}{\hat{\sigma}_1^2} = \frac{\text{SNR}_T}{\text{SNR}_t}\frac{\alpha_t}{\alpha_T}$$

This equality holds if we define the underlying diffusion schedule such that the SNR ratio matches our variance schedule. Furthermore, for a standard VP-SDE where $\alpha_t^2 + \sigma_t^2 = 1$, we typically have $\alpha_T \approx 0$ (so $\text{SNR}_T \approx 0$) and $\alpha_0 = 1$.

By simplifying the general coefficients under the condition that our bridge schedule is consistent with the underlying diffusion physics, we recover the forms:

$$a_t = \frac{\bar{\sigma}_t^2}{\sigma_1^2}, \quad b_t = \frac{\sigma_t^2}{\sigma_1^2}, \quad c_t = \frac{\bar{\sigma}_t \sigma_t}{\sigma_1}. \tag{23}$$

where $\bar{\sigma}_t^2 = \sigma_1^2 - \sigma_t^2$. Specifically, notice that at endpoints:

$t = 0(\text{Target}) \implies \sigma_0 = 0 \implies a_0 = 1, b_0 = 0, c_0 = 0$, recovering $\mathbf{z}_0$.

$t = 1(\text{Source}) \implies \sigma_1 = \sigma_1 \implies a_1 = 0, b_1 = 1, c_1 = 0$, recovering $\mathbf{z}_T$.

Thus, Eq. 10 in the main text is theoretically grounded in the DDBM framework, providing a specific, tractable parameterization for video removal.

### B.2. Derivation of the Guidance Term $h$

In the SDE formulation (Eq. 7 in main text), we introduced the guidance term $h(\mathbf{z}_t, t, \mathbf{z}_{\text{src}}) = \nabla_{\mathbf{z}_t} \log p(\mathbf{z}_{\text{src}}|\mathbf{z}_t)$. Since $p_{\text{diff}}(\mathbf{z}_{\text{src}}|\mathbf{z}_t)$ is Gaussian (as shown above), we can derive $h$ analytically.

Recall that:

$$p_{\text{diff}}(\mathbf{z}_{\text{src}}|\mathbf{z}_t) = \mathcal{N}\left(\mathbf{z}_{\text{src}}; \frac{\alpha_1}{\alpha_t}\mathbf{z}_t, \Sigma_{1|t}\mathbf{I}\right)$$

where $\Sigma_{1|t} = \sigma_1^2 - \frac{\alpha_1^2}{\alpha_t^2}\sigma_t^2$.

Taking the logarithm of the Gaussian density:

$$\log p(\mathbf{z}_{\text{src}}|\mathbf{z}_t) = -\frac{\left\|\mathbf{z}_{\text{src}} - \frac{\alpha_1}{\alpha_t}\mathbf{z}_t\right\|^2}{2\Sigma_{1|t}} + C$$

Computing the gradient with respect to $\mathbf{z}_t$:

$$h(\mathbf{z}_t, t, \mathbf{z}_{\text{src}}) = \nabla_{\mathbf{z}_t}\left(-\frac{\left\|\mathbf{z}_{\text{src}} - \frac{\alpha_1}{\alpha_t}\mathbf{z}_t\right\|^2}{2\Sigma_{1|t}}\right)$$

$$= \frac{\frac{\alpha_1}{\alpha_t}}{\Sigma_{1|t}}\left(\mathbf{z}_{\text{src}} - \frac{\alpha_1}{\alpha_t}\mathbf{z}_t\right)$$

This explicit form confirms that the guidance term required for the SDE formulation is indeed analytically tractable.

The coefficients expressions given in Eq. 10 of the main text ($a_t = \bar{\sigma}_t^2/\sigma_1^2$, etc.) are instances of the general form derived above under a specific variance schedule. Specifically, when the bridge variance schedule satisfies $\text{SNR}_t/\text{SNR}_1 = \bar{\sigma}_t^2/\sigma_t^2$, the general formulas simplify to the specific forms used in our implementation. This demonstrates that our method strictly adheres to the DDBM theoretical framework.

### B.3. Equivalence of Training Objective

In the main text, we define the training objective as a velocity matching loss:

$$\mathcal{L} = \mathbb{E}\left[\|\mathbf{v}_\theta - \mathbf{u}_t\|^2\right]$$

where the target velocity is given by $\mathbf{u}_t = \frac{d\mathbf{z}_t}{dt}$. Here, we prove that this objective is theoretically equivalent to Denoising Bridge Score Matching.

The true score function of the bridge process is defined as $\nabla_{\mathbf{z}_t}\log p_t(\mathbf{z}_t|\mathbf{z}_{\text{tgt}}, \mathbf{z}_{\text{src}})$ (Zhou et al., 2023a). Using the Gaussian conclusion from Appendix B.1, we can explicitly write:

$$\nabla_{\mathbf{z}_t}\log p_t = -\frac{\mathbf{z}_t - \boldsymbol{\mu}_t}{\sigma_{t|\text{bridge}}^2} = -\frac{\mathbf{z}_t - (a_t\mathbf{z}_{\text{tgt}} + b_t\mathbf{z}_{\text{src}})}{\sigma_{t|\text{bridge}}^2}$$

Let us define the normalized residual term $\boldsymbol{\epsilon}_{\text{res}} = \frac{\mathbf{z}_t - a_t\mathbf{z}_{\text{tgt}} - b_t\mathbf{z}_{\text{src}}}{c_t}$, where $c_t = \sigma_{t|\text{bridge}}$.

If we parameterize the model to predict this normalized residual (which is equivalent to predicting velocity under the V-Prediction framework), i.e., letting the model output $\mathbf{v}_\theta \propto \boldsymbol{\epsilon}_{\text{res}}$, then the Score Matching loss:

$$\mathcal{L}_{\text{SM}} = \mathbb{E}\left[\|\mathbf{s}_\theta - \nabla_{\mathbf{z}_t}\log p_t\|^2\right]$$

can be rewritten in the Mean Squared Error (MSE) form:

$$\mathcal{L}_{\text{SM}} \propto \mathbb{E}\left[\left\|\mathbf{v}_\theta(\mathbf{z}_t) - \underbrace{\left(\frac{a_t}{\rho_t}\boldsymbol{\epsilon} - \frac{c_t}{\rho_t}\mathbf{z}_{\text{tgt}}\right)}_{\text{Ground Truth Velocity } \mathbf{u}_t}\right\|^2\right]$$

This transformation utilizes the reparameterization formula $\mathbf{z}_t = a_t\mathbf{z}_{\text{tgt}} + b_t\mathbf{z}_{\text{src}} + c_t\boldsymbol{\epsilon}$.

Time

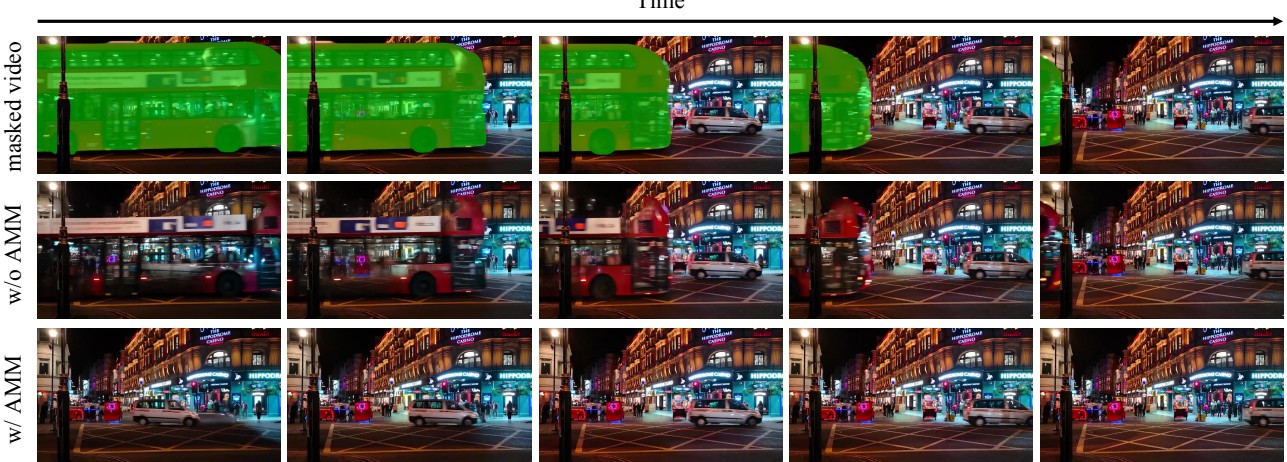

*Figure 4.* Comparison of w/ and w/o adaptive mask modulation.

Minimizing our proposed velocity regression loss $\mathcal{L}$ is mathematically equivalent to minimizing the Score Matching error of the bridge process. This ensures that the vector field learned by the model can correctly push the distribution from the source video to the target video. This proves that our MSE training objective is mathematically grounded in Score Matching theory, guaranteeing that the model learns the correct gradient field for the bridge process.

## C. Ablation Study

### C.1. Influence of AMM

We conduct an ablation study to evaluate the impact of AMM. As shown in Figure 4, a bus occupies approximately 80 percent of the video frame. When AMM is disabled, our BridgeRemoval fails to fully erase the large object. The reconstructed region retains visible artifacts and structural remnants of the bus, indicating that the strong structural prior from the source video impedes sufficient deviation needed for plausible inpainting of extensive occlusions. In contrast, with AMM enabled, the model successfully removes the entire bus and synthesizes a coherent background that aligns with the surrounding scene context and maintains temporal consistency across frames. This demonstrates that AMM effectively mitigates the rigidity induced by the bridge prior. By amplifying the embeddings in unmasked background regions, AMM enhances the model's reliance on reliable contextual cues while granting greater generative flexibility within the masked area.

### C.2. Influence of Inference Steps

Diffusion-based video object removal methods (Miao et al., 2025; Jiang et al., 2025; Lee et al., 2025) typically initialize inference from random noise and often require 50 or more denoising steps. In contrast, our BridgeRemoval starts inference directly from the source video. To investigate the impact of inference steps, we evaluate our method on the DAVIS dataset using 10, 20, 30, 40, and 50 steps, as reported in Table 4. The results show that with only 10 steps, BridgeRemoval already achieves the best CLIP-F, PSNR, and MSE. As the number of inference steps increases, CLIP-T gradually improves while PSNR steadily decreases. This trend is intuitive: more steps lead to cleaner object removal, reflected in higher CLIP-T, but at the cost of background fidelity, resulting in lower PSNR.

Figure 5 illustrates this behavior on a representative example. At 10 steps, the masked region lacks fine texture details. As the step count increases, the inpainted content becomes progressively more realistic and coherent with the scene semantics. To balance object removal quality and background preservation, we adopt 50 inference steps in our main experiments.

## D. More Quantitative Results

Some methods, such as ROSE (Miao et al., 2025), employ VBench (Huang et al., 2024) to evaluate the performance of video object removal. However, this may be inappropriate. VBench was specifically designed for assessing video generation models, decomposing video generation quality into multiple dimensions including Motion Smoothness, Background

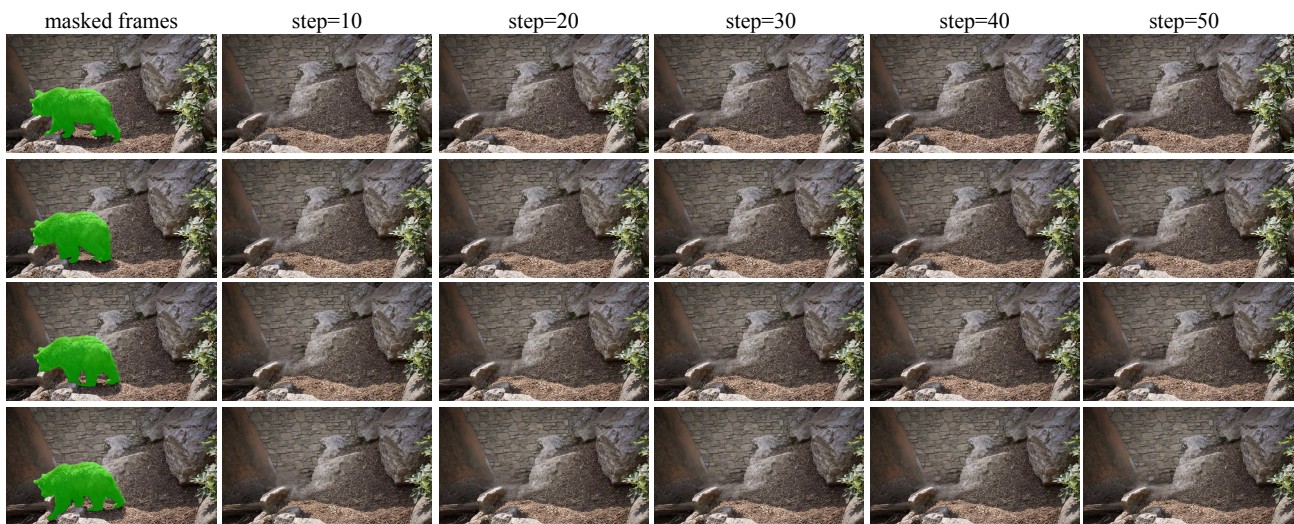

*Figure 5.* Visualization results of BridgeRemoval at different inference steps. This video is from the DAVIS dataset.

*Table 4.* Comparison of different inference steps on DAVIS dataset. **Red** stands for the best.

| | Video Quality | | Unmasked Region Preservation | | |
|---|---|---|---|---|---|
| Inference steps | CLIP-T ↑ | CLIP-F ↑ | PSNR ↑ | MSE ↓ | LPIPS ↓ |
| 10 | 0.2765 | **0.9771** | **28.4155** | **128.6873** | 0.0985 |
| 20 | 0.2779 | 0.9767 | 28.3085 | 131.6830 | 0.0977 |
| 30 | 0.2784 | 0.9769 | 28.2897 | 132.2086 | 0.0975 |
| 40 | 0.2786 | 0.9767 | 28.2555 | 132.7415 | **0.0973** |
| 50 | **0.2788** | 0.9768 | 28.2154 | 133.3605 | 0.0977 |

*Table 5.* VBench-based evaluation on DAVIS dataset. **Red** stands for the best, **Blue** stands for the second best.

| Method | VBench | | | | |
|---|---|---|---|---|---|
| | Motion Smoothness ↑ | Background Consistency ↑ | Temporal Flickering ↓ | Subject Consistency ↑ | Imaging Quality ↑ |
| Senorita (Zi et al., 2025b) | 0.9724 | 0.8885 | 0.9437 | 0.8152 | 0.5187 |
| Ditto (Bai et al., 2025a) | 0.9748 | 0.9166 | **0.9345** | **0.9119** | **0.6948** |
| ICVE (Liao et al., 2025) | **0.9803** | 0.9173 | 0.9445 | 0.8983 | 0.5325 |
| ROSE (Miao et al., 2025) | 0.9750 | **0.9245** | 0.9384 | 0.9051 | 0.6239 |
| VACE (Jiang et al., 2025) | 0.9716 | 0.9173 | **0.9329** | 0.8974 | **0.6618** |
| GenOmnimatte (Lee et al., 2025) | **0.9757** | 0.9216 | 0.9384 | **0.9119** | 0.6179 |
| Ours | 0.9736 | **0.9305** | 0.9358 | **0.9111** | 0.6376 |

*Table 6.* VBench-based evaluation on BridgeRemoval-Bench. **Red** stands for the best, **Blue** stands for the second best.

| Method | VBench | | | | |
|---|---|---|---|---|---|
| | Motion Smoothness ↑ | Background Consistency ↑ | Temporal Flickering ↓ | Subject Consistency ↑ | Imaging Quality ↑ |
| Senorita (Zi et al., 2025b) | 0.9901 | 0.9162 | 0.9770 | 0.8771 | 0.5470 |
| Ditto (Bai et al., 2025a) | 0.9892 | 0.9479 | **0.9715** | **0.9589** | **0.7423** |
| ICVE (Liao et al., 2025) | **0.9935** | 0.9402 | 0.9791 | **0.9561** | 0.5803 |
| ROSE (Miao et al., 2025) | 0.9918 | 0.9470 | 0.9764 | 0.9536 | 0.6257 |
| VACE (Jiang et al., 2025) | 0.9907 | **0.9532** | **0.9740** | 0.9436 | **0.6462** |
| GenOmnimatte (Lee et al., 2025) | **0.9925** | 0.9417 | 0.9769 | 0.9558 | 0.6373 |
| Ours | 0.9924 | **0.9554** | 0.9758 | 0.9557 | 0.6429 |

| masked frames | ROSE | VACE | GenOmnimatte | Ours |
|---|---|---|---|---|

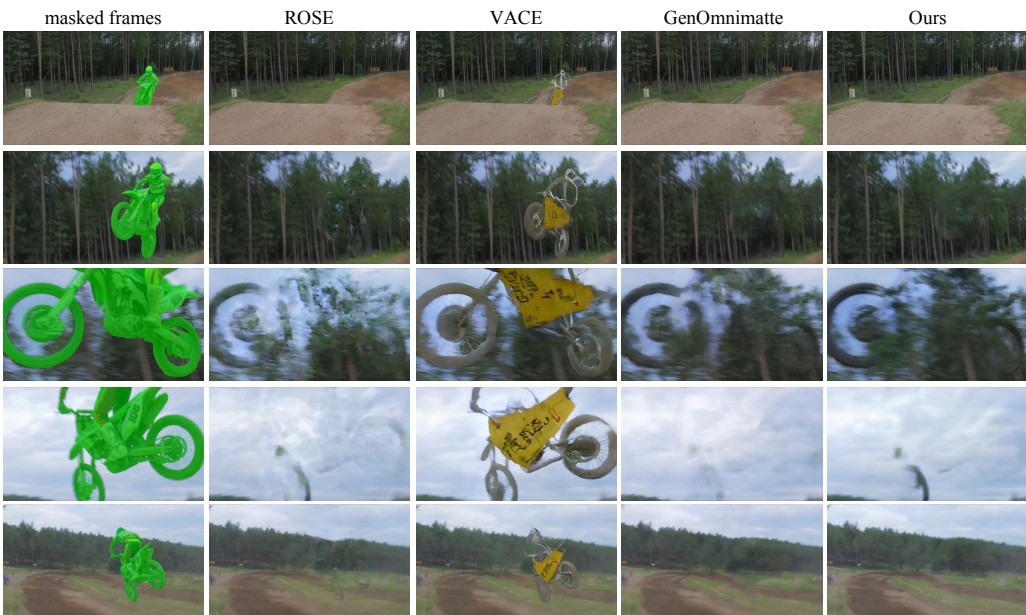

*Figure 6.* Limitations in removing fast moving objects. This video is from the DAVIS dataset.

Consistency, Temporal Flickering, Subject Consistency, and Imaging Quality. These metrics have limitations when applied to the video object removal task because they cannot determine whether the target object has been genuinely removed. For instance, VACE (Jiang et al., 2025) often generates a new object that conforms to the shape of the mask. As shown in Tables 5 and 6, VACE achieves relatively high scores on Temporal Flickering and Imaging Quality, yet this does not indicate effective object removal. Furthermore, the scores across different methods on VBench metrics exhibit very low discriminability, which fails to reflect meaningful performance gaps. For example, all methods achieve Motion Smoothness scores above 0.97, suggesting limited utility of this metric for distinguishing object removal quality.

## E. Limitation

One limitation of BridgeRemoval lies in its performance on videos containing objects with rapid motion. As illustrated in Figure 6, when the target object moves quickly across frames, our method may not fully remove it and can leave visible artifacts or ghosting effects in the inpainted region. Importantly, this difficulty is not specific to our approach but appears to be a general challenge in the field of video object removal. We observe that state of the art methods such as ROSE (Miao et al., 2025), VACE (Jiang et al., 2025), and GenOmnimatte (Lee et al., 2025) also exhibit similar failure modes on the same types of fast moving objects. In future work, we plan to explore hybrid architectures that integrate explicit motion estimation with stochastic bridge models to enhance robustness in high speed scenarios.

## F. More Qualitative Results

More qualitative samples can be visited at: `https://bridgeremoval.github.io/`

