# OpenReview forum: "Learning Stochastic Bridges for Video Object Removal via Video-to-Video Translation"
_ICML.cc/2026/Conference — ICML 2026 regular_

### Official Review · Reviewer_zjdA · 2026-03-08

**Soundness:** 3
**Presentation:** 3
**Significance:** 2
**Originality:** 2
**Overall Recommendation:** 4
**Confidence:** 3

**Summary:**

This paper proposes a novel video object removal framework named BridgeRemoval. Most existing diffusion-based methods typically adopt a "noise-to-data" generation paradigm, which often discards the rich structural and contextual priors present in the original input video. To address this issue, the authors reformulate the task as a "video-to-video" translation problem, utilizing a Variance-Preserving Stochastic Differential Equation (VP-SDE) to construct a direct stochastic path from the source video to the target video. To overcome the limitation where a strong source video prior hinders the removal of large objects, the paper introduces an Adaptive Mask Modulation (AMM) strategy, which dynamically adjusts feature embeddings based on mask characteristics. Finally, the authors construct a new benchmark dataset, BridgeRemoval-Bench, comprising 150 high-quality videos. Experimental results demonstrate that this method significantly outperforms existing state-of-the-art methods in both visual quality and temporal consistency.

**Compliance With Llm Reviewing Policy:**

Affirmed.

**Final Justification:**

Most of my concerns have been addressed.

**Key Questions For Authors:**

1.	What are the specific, significant advantages of the VP-SDE bridge for the task presented in this paper?

2.	Given that both the bridge model and feature modulation are direct applications of existing techniques, could you further clarify the exact core original contribution of this paper to the computer vision or generative modeling communities?

**Limitations:**

yes

**Strengths And Weaknesses:**

Strengths:

1. The paper addresses a video editing problem with practical application value.

2. It demonstrates solid quantitative metrics and visual results on the proposed BridgeRemoval-Bench dataset.

3. The overall writing is fluent, and the algorithmic pseudo-codes provided in the appendix are highly helpful for understanding the model pipeline.

Weakness:

1.	SDE-based bridge models have been widely applied in areas such as image-to-image translation, and AMM is essentially a basic spatial feature affine transformation and modulation. Simply combining these two existing technologies and applying them to the video object removal task represents a highly incremental engineering effort.

2.	The paper claims to utilize a variance-preserving VP-SDE bridge formulation instead of a standard Brownian bridge. However, the authors fail to demonstrate—through comprehensive ablation studies or rigorous theoretical comparisons—what specific, irreplaceable advantages the VP-SDE offers over other bridge formulations for the specific task of "video object removal".

3.	The current framework is deeply coupled with and completely relies on a frozen Wan2.1-1.3B pre-trained VAE. Although the baselines include a method based on CogVideoX-5B (i.e., Senoria), the authors did not deploy or test their proposed BridgeRemoval framework on other mainstream video generation architectures like CogVideo.

---

> ### Author Rebuttal · Authors · 2026-03-29
>
> We sincerely appreciate your recognition of the practical value of our video editing task, the solid quantitative and qualitative results. We are happy to engage in a thorough discussion and hope it will address your concerns and questions.
>
> ## Key Differences from DDBM (**core original contribution**)
> We sincerely thank the reviewer for pointing out the connection between our method and prior SDE-based bridge models such as DDBM. We respectfully clarify that our method is not a simple task-level application of vanilla DDBM. Rather, starting from the original DDBM framework, we further derive the interpolation, GT target, and reverse sampling rule used in our paper for video object removal.
>
> ### 1. On the interpolation coefficient derivation
> Vanilla DDBM derives a general Gaussian bridge with coefficients $(A_t,B_t,C_t)$. However, under the DDBM theory, these coefficients are expressed through coupled SNR ratios, so the bridge marginal remains in a fairly complicated general form rather than a simple interpolation directly usable for source-to-target video removal.
>
> We introduce a variance-driven parameterization (see **Appendix B.1**) and show that, when the variance schedule is aligned with the SNR ratios, the general DDBM bridge collapses to the tractable ternary interpolation used in our main text, with coefficients $(a_t,b_t,c_t)$. Thus, our interpolation correctly preserves the boundary conditions and makes forward bridge sampling practical for our setting.
>
> ### 2. On the GT target derivation
> In DDBM, the original supervision is the conditional bridge score, and in practice it is implemented through a covariance-dependent target. Therefore, the DDBM GT remains tied to the original score formulation and bridge covariance quantities. In our setting, we instead require a GT that is directly compatible with our ternary bridge parameterization and velocity-driven objective.
>
> **Appendix B.3** shows that $u_t$ is analytically derived from our ternary interpolation, together with the normalized residual form, and that minimizing our velocity regression loss is mathematically equivalent to denoising bridge score matching. Therefore, compared with DDBM, the difference is not a cosmetic reparameterization, but a new GT derivation tailored to our bridge.
>
> ### 3. On the reverse sampling derivation
> In DDBM, the reverse process is written in score/guidance form rather than around an explicit recovered target latent. In our formulation, once the model predicts the bridge velocity, we analytically invert this relation to recover the target estimate $\hat{z}_{0|t}$ and then use it in the reverse solver. We briefly show the key derivation here for clarity. Starting from
>
> $
> z_t = a_t z_{\text{tgt}} + b_t z_{\text{src}} + c_t \epsilon,
> \qquad
> u_t = \frac{a_t}{\rho_t}\epsilon - \frac{c_t}{\rho_t} z_{\text{tgt}},
> $
>
> we have
> $
> \rho_t u_t = a_t \epsilon - c_t z_{\text{tgt}}.
> $
>
> Combining this with
> $
> z_t - b_t z_{\text{src}} = a_t z_{\text{tgt}} + c_t \epsilon,
> $
>
> gives
> $
> a_t (z_t - b_t z_{\text{src}}) - c_t \rho_t u_t = \rho_t^2 z_{\text{tgt}}.
> $
>
> Hence
> $
> z_{\text{tgt}}=\frac{1}{\rho\_t^2/c\_t}\left(\frac{a\_t}{c\_t}z\_t-\frac{a\_t b\_t}{c\_t}z\_{\text{src}}-\rho\_t u\_t\right),
> $
> and replacing $u_t$ by $\hat v_t$ yields the recovery formula for $\hat z_{0|t}$ in the main text.
>
> Our sampler is built around analytic recovery of $\hat z_{0|t}$ and then a bridge-consistent reverse update, rather than directly following the vanilla DDBM score/guidance form. This analytic inversion is exactly what connects our velocity parameterization to generative inference.
>
> ## On the Motivation and Novelty of AMM
> Due to the character limit, we kindly invite the reviewer to refer to our response to Reviewer xY1E under **On the Motivation and Novelty of AMM**.
>
> ## Comparison with Bridge Matching
> Due to the character limit, we kindly invite the reviewer to refer to our response to Reviewer Rxds under **Comparison with Bridge Matching**.
>
> ## Additional Results on Another Video Generation Backbone
> We deployed BridgeRemoval on another video generation model, CogvideoX-5B. This variant achieves competitive results, including the highest CLIP-T score of 0.3001. However, due to the much larger model size, its inference speed is slower. For this reason, we choose Wan2.1-1.3B as our backbone, which offers a better balance between performance and efficiency.
> |BridgeRemoval-Bench|CLIP-T ↑|CLIP-F ↑|PSNR ↑|MSE ↓|LPIPS ↓|Runtime (s/frame)↓|
> |---|---:|---:|---:|---:|---:|---:|
> |CogvideoX-5B+SDE Bridge+AMM| 0.3001| 0.9884|32.1522|53.7748|0.0479|2.043|
> |Wan2.1-1.3B+SDE Bridge+AMM (BridgeRemoval)|0.2922|0.9917|33.2394|46.3525|0.0415|1.111 |
>
> ## On the Advantages of the VP-SDE Bridge for Video Object Removal
> Extensive results suggest that, for our task, the main advantage of the VP-SDE bridge is that it achieves strong background preservation compared with diffusion-based formulations, while also maintaining better object removal capability than the ODE bridge.

---

> > ### Author Rebuttal · Reviewer_zjdA · 2026-04-03
> >
> > This response successfully clarifies my questions regarding the AMM module. Furthermore, the detailed mathematical derivations for the ternary interpolation and analytical inversion fully resolve my theoretical concerns. Finally, the new experiments on the larger CogvideoX-5B backbone clearly demonstrate the model's robustness. I will raise my score accordingly.

---

> > > ### Author Response · Authors · 2026-04-03
> > >
> > > We would like to express our sincere gratitude for your positive comments on our rebuttal. We are pleased that the provided clarifications, mathematical derivations, and expanded experiments fully resolved your concerns. Thank you again for your valuable insights and for adjusting your score in support of our work.

---

### Official Review · Reviewer_Rxds · 2026-03-09

**Soundness:** 4
**Presentation:** 4
**Significance:** 4
**Originality:** 3
**Overall Recommendation:** 3
**Confidence:** 4

**Summary:**

# Summary

This paper propose a novel framework BridgeRemoval that reformulates video object removal as a video-to-video translation task using a stochastic bridge model. BridgeRemoval establishes a direct stochastic path from the source video distribution to the target distribution that leverages the input video as a strong prior to ensure spatial coherence and temporal consistency. To handle the trade-off where strong priors might hinder the removal of large objects, the authors introduce an Adaptive Mask Modulation (AMM) strategy that dynamically adjusts the influence of the input embeddings based on mask characteristics.

The key contributions can be summarized as follows:

1. A novel video-to-video generative model that creates a direct probability path between source and target videos together with a new video object removal benchmark.

2. AMM mechanism balances background fidelity and generative flexibility by dynamically modulating features in masked and unmasked regions.

3. Extensive experiments demonstrating that BridgeRemoval achieves SOTA performance.

**Compliance With Llm Reviewing Policy:**

Affirmed.

**Key Questions For Authors:**

# Key questions:

1. The results illustrated in the paper figures mainly focus on single object removal. Can the model achieve satisfactory performance when there are multiple objects or large objects to remove?

2. The method uses a stochastic bridge. Have the authors considered or experimented with deterministic flow matching or rectified flows for this task? Would a deterministic path offer faster inference or more stable results for object removal, given the strong prior?

3. The proposed method is good at perserving background, especially static background. Can the model deal with situations where the background also changes, e.g. view point change or even different shots?

**Limitations:**

please refer to weaknesses and questions.

**Strengths And Weaknesses:**

# Strengths

1. The paper is well-written with a clear structure and the experiments are extensive.

2. The shift from noise-to-data to video-to-video translation via stochastic bridges is a major strength.

3. A new benchmark BridgeRemoval-Bench is proposed for video object removal task.


# Weaknesses

1. The framework directly adapts DDBM structure to video object removal task with relatively little theoretical improvement. This makes the contribution relatively marginal.

2. The BridgeRemoval framwork requires object mask of every frame. In real-world scenarios, obtaining such masks often requires manual annotation or separate tracking/segmentation models which is either expensive or can be imperfect and noisy.

3. As described in the paper, the method exhibits noticeable failures when removing fast-moving objects, resulting in ghosting or incomplete removal. And the model requires high computational and time cost as reported 1.111s per frame.

---

> ### Author Rebuttal · Authors · 2026-03-29
>
> We sincerely appreciate your acknowledgement of our paper as "well-written with a clear structure", our stochastic bridge formulation for video-to-video translation as "a major strength", and our proposed benchmark "BridgeRemoval-Bench" for the video object removal task. We are happy to engage in a thorough discussion and hope it will address your concerns and questions.
>
> ## Key Differences from DDBM
> We respectfully clarify that our method is not a simple task-level application of vanilla DDBM. Rather, starting from the original DDBM framework, we further derive the interpolation, GT target, and reverse sampling rule used in our paper for video object removal. Due to the character limit, we kindly invite the reviewer to refer to our response to Reviewer zjdA under Key Differences from DDBM, where we explain these differences in detail.
>
> ## Additional Results on Videos with Multiple Objects, Large Objects, and Dynamic Backgrounds
> We thank the reviewer for this valuable suggestion and address this concern with additional experiments. In fact, BridgeRemoval-Bench is already diverse and contains videos with multiple objects, large objects, and dynamic backgrounds. We further extract these three subsets and construct dedicated test sets. Due to the character limit, we provide the quantitative results in https://bridgeremoval.github.io/rebuttal/ (**Table 10, 11, 12**).
> The results show that our method still achieves the best overall performance across these challenging scenarios.
>
> ## Comparison with Bridge Matching
>
> We also compared our method with different bridging strategies. Specifically, we implemented a deterministic ODE bridge following the design of LBM [1], which constructs a linear probability path between the source latent $z_{src}$ and the target latent $z_{tgt}$ as:
>
> $
>     z_t = (1-t)z_{src} + t z_{tgt}, \quad t \in [0, 1]
> $
>
> The model is trained to predict the constant velocity $v_t = \frac{dz_t}{dt} = z_{tgt} - z_{src}$, a formulation that can be viewed as a latent-space variant of Flow Matching or Rectified Flow.
>
> This variant requires only 4 inference steps and is therefore very efficient. However, although it preserves the background reasonably well, its object removal performance is weaker. One possible explanation is that the ODE formulation is deterministic. When multiple plausible fillings exist for the masked region, it may tend to produce an averaged solution, which leads to blurrier results. Visual comparisons are provided in https://bridgeremoval.github.io/rebuttal/ (see **Figure 14, 15**}.
>
> | DAVIS-dataset | CLIP-T ↑ | CLIP-F ↑ | PSNR ↑ | MSE ↓ | LPIPS ↓ |Runtime (s/frame) ↓|
> |---|---:|---:|---:|---:|---:|---:|
> | Wan2.1-1.3B+ODE Bridge+AMM |0.2589 | 0.9594 | 28.0010 | 147.7071 | 0.1006 |0.267|
> | Wan2.1-1.3B+SDE Bridge+AMM (BridgeRemoval) | 0.2788 | 0.9768 | 28.2154 | 133.3605| 0.0977 |1.111|
>
> | BridgeRemoval-Bench | CLIP-T ↑ | CLIP-F ↑ | PSNR ↑ | MSE ↓ | LPIPS ↓ |Runtime (s/frame) ↓|
> |---|---:|---:|---:|---:|---:|---:|
> | Wan2.1-1.3B+ODE Bridge+AMM | 0.2733 | 0.9841 | 32.6181 | 45.1564 | 0.0498 |0.267|
> | Wan2.1-1.3B+SDE Bridge+AMM (BridgeRemoval) | 0.2922 | 0.9917 | 33.2394 | 46.3525 | 0.0415 |1.111|
>
> ## On the Requirement of Frame-Wise Masks
>
> We appreciate this concern. In practice, current mask-free video object removal methods still struggle to remove objects precisely. By contrast, existing mask-based methods such as ROSE, VACE, GenOmnimatte, and our BridgeRemoval all rely on frame-wise masks. Such masks can now be constructed in a largely automated way using modern segmentation and tracking tools such as SAM-2, which substantially reduces manual annotation cost. We would also like to clarify that our method is not overly sensitive to imperfect masks. In real applications, the available masks are often coarse and noisy rather than pixel-accurate. Accordingly, training with similarly coarse masks can improve robustness and generalization. Some examples on our project page show that the model can even remove shadows jointly with the target object when the shadows are not annotated in the mask.
>
> ## On the Limitation in Removing Fast-Moving Objects
>
> We agree that removing fast-moving objects remains challenging. This is a common difficulty for current video object removal methods, including ROSE, VACE, and GenOmnimatte, which also show artifacts or incomplete removal in such cases. Our method does not fully solve this problem either, but it still achieves the best overall performance among existing approaches.
>
> ## Regarding Inference Speed
>
> We would like to clarify that our model has a relatively small computational cost, with only 1.3B parameters, which is the smallest among all compared methods. Its inference speed is 1.111 seconds per frame, ranking second among all methods in our comparison.
>
> Reference
>
> [1] Chadebec, Clément, et al. "LBM: Latent bridge matching for fast image-to-image translation." Proceedings of the IEEE/CVF International Conference on Computer Vision. 2025.

---

> > ### Author Rebuttal · Reviewer_Rxds · 2026-04-03
> >
> > Thanks for the response. I think the authors have addressed most of my concerns and I prefer to accept this paper now.

---

> > > ### Author Response · Authors · 2026-04-06
> > >
> > > We would like to thank the Reviewer for the confirmation that the concerns have been fully resolved. We appreciate the reviewer's positive feedback and support for our work. We hope our clarifications have resolved the issues you raised. If so, we would be grateful if you would consider increasing the score.

---

### Official Review · Reviewer_MGpV · 2026-03-10

**Soundness:** 3
**Presentation:** 1
**Significance:** 2
**Originality:** 2
**Overall Recommendation:** 4
**Confidence:** 5

**Summary:**

This paper proposes BridgeRemoval, which reformulates video object removal as a video-to-video translation task based on a stochastic bridge. Instead of starting generation from pure Gaussian noise, the method builds a VP-SDE bridge between the source video (with objects) and the target video (with objects removed). In this way, the input video is used as a strong prior in the generation process. With this formulation, the model can better preserve background structure and temporal information, while only modifying the regions related to the object.

**Compliance With Llm Reviewing Policy:**

Affirmed.

**Final Justification:**

The authors have addressed most of my concerns. However, the novelty is limited, which is also pointed out by other reviewers. I prefer to give a weak accept.

**Key Questions For Authors:**

1.	Can the core advantage of the bridge formulation over standard diffusion be quantified more clearly?
2.	Is AMM better than simpler mask conditioning methods?
3.	Could the ratio of synthetic videos, social media videos, and stock videos in the benchmark affect the conclusions?
4.	Can this method be easily extended to other video editing tasks?

**Limitations:**

Besides the limitation mentioned by the authors in the appendix (videos containing objects with rapid motion), the method also has the following limitations:
1. Slow inference speed. The model runs at the normal inference speed of a 1.3B model and cannot achieve real-time performance.
2. Training requires paired data. The method needs triplets of source video, clean target video, and mask, which are difficult to collect at large scale in real-world scenarios. Although the authors construct such data using a synthetic/composite pipeline, the method still heavily depends on the quality of supervised data.
3. Performance improvement may partly depend on the backbone and training details. The method is built on the pretrained Wan2.1-1.3B model. Therefore, it is difficult to clearly separate how much of the improvement comes from the bridge formulation and how much comes from the backbone capacity, training strategy, or data construction.

**Strengths And Weaknesses:**

Strength

1.  The task formulation is natural and well motivated.

2.  The method is complete and the technical pipeline is clearly described.

3.  The paper introduces a new benchmark for this task.

Weaknesses

1.	Although the main idea is reasonable, the method mainly adapts existing bridge modeling frameworks to the video object removal task, together with mask modulation. It does not introduce a fundamentally new generative modeling framework. Therefore, the novelty mainly comes from task reformulation and engineering integration, while the theoretical innovation is limited.

2.	The proposed AMM module is conceptually similar to mask-aware feature modulation or FiLM-style conditioning. Although it is useful for the task, the module itself is not a strong technical innovation.

3.	Some important ablation studies are missing:
(1) How much improvement does the bridge formulation bring compared to standard diffusion?
(2) How much improvement does AMM bring compared to simple mask concatenation?

4.	Metrics such as CLIP-T, CLIP-F, PSNR, and LPIPS provide partial evaluation, but they may not accurately measure whether the object is truly removed and whether the video is temporally consistent. It would be helpful to include additional metrics such as FVD, VFID, tLPIPS, and optical-flow consistency.

5. The authors use the wrong anonymized github address, which violates the anonymized rule.

---

> ### Author Rebuttal · Authors · 2026-03-29
>
> We are grateful for your recognition of our method as "natural and well motivated" and our technical pipeline as "complete and clearly described". We welcome the opportunity to elaborate further and are confident that the following discussion will adequately address your concerns and questions.
>
> ## Key Differences from DDBM
> We would like to gently point out that the innovation of our work is not limited to task reformulation and engineering integration, and BridgeRemoval is a novel bridge-based framework specifically designed for video object removal. Due to the character limit, we kindly invite the reviewer to refer to our response to Reviewer zjdA under **Key Differences from DDBM**, where we explain in detail the theoretical differences between our framework and the original DDBM formulation.
>
> ## On the Motivation and Novelty of AMM
> Due to the character limit, we kindly invite the reviewer to refer to our response to Reviewer xY1E under **On the Motivation and Novelty of AMM**, where we clarify the role, motivation, and technical rationale of AMM in more detail.
>
> ## Ablation on the Contributions of the Bridge and AMM
> We thank the reviewer for this valuable suggestion and have provided the experimental results for this ablation. Due to the character limit, we kindly invite the reviewer to refer to our response to Reviewer xY1E under **Ablation on the Contributions of the Bridge and AMM**, where we report the corresponding comparisons in detail.
>
> | Method | CLIP-T ↑ | CLIP-F ↑ | PSNR ↑ | MSE ↓ | LPIPS ↓ |
> |---|---:|---:|---:|---:|---:|
> | Wan2.1-1.3B | 0.2810 | 0.9815 | 30.7283 | 68.2547 | 0.0453 |
> | Wan2.1-1.3B+SDE Bridge | 0.2846 | 0.9880 | 32.9215 | 48.6971 | 0.0439 |
> | Wan2.1-1.3B+SDE Bridge+AMM (BridgeRemoval) | **0.2922** | **0.9917** | **33.2394** | **46.3525** | **0.0415**|
>
> ## Results by Data Source: Synthetic, Social Media, and Stock Videos
> To address this concern, we separately evaluate the three subsets in the benchmark, namely stock videos, social media videos, and synthetic videos BridgeRemoval still shows overall stronger object removal capability and better background preservation than prior methods. Due to the character limit, we provide the quantitative results in https://bridgeremoval.github.io/rebuttal/ (See **Tables 7, 8, 9**).
>
> ## Regarding Inference Speed
> We agree that the current inference speed does not yet reach real-time performance. However, this is a common limitation of generation-based video object removal methods in general. Our method runs at 1.111 s/frame, which ranks second among all compared methods, while maintaining strong removal quality.
>
> ## On the Requirement of Paired Training Data
> We appreciate this concern. In practice, current mask-free video object removal methods still struggle to remove objects precisely. By contrast, mask-based methods such as ROSE, VACE, GenOmnimatte and BridgeRemoval all rely on source video, clean target video, and mask triplets. Such paired data can now be constructed in a largely automated way using tools such as SAM-2.
>
> We also would like to clarify that our method is not overly sensitive to supervision quality. In real applications, the available masks are often coarse rather than perfect. Accordingly, training with similarly coarse masks can improve robustness and generalization. Some examples on our project page show that the model can even remove shadows jointly with the target object when the shadows are not explicitly annotated in the mask.
>
> ## Additional Evaluation Metrics
> We appreciate this suggestion. FVD and VFID are commonly used for video generation tasks to measure generation quality, but they are less suitable for video editing. Therefore, we additionally evaluate tLPIPS and GWE（Global Warp Error）. As shown in the table, BridgeRemoval achieves the best tLPIPS and the second-best GWE. This further demonstrates that BridgeRemoval achieves both effective object removal and strong temporal consistency.
>
> | Method | DAVIS-dataset GWE ↓ | DAVIS-dataset tLPIPS ↓ | BridgeRemoval-Bench GWE ↓ | BridgeRemoval-Bench tLPIPS ↓ |
> |---|---:|---:|---:|---:|
> | senorita | 0.1887 | 0.1679 | 0.1800 | 0.0597 |
> | Ditto | 0.2466 | 0.1536 | 0.2113 | 0.0552 |
> | ICVE | 0.1944 | 0.1616 | 0.1791 | 0.0513 |
> | ROSE | 0.1702 | 0.1698 | 0.1722 | 0.0561 |
> | VACE | 0.1680 | 0.1776 | 0.1732 | 0.0618 |
> | GenOmnimatte | 0.1744 | 0.1711 | 0.1735 | 0.0552 |
> | Ours | **0.1661** | **0.1489** | **0.1650** | **0.0509** |
>
> ## Clarification on the Anonymized Project Link
> **We respectfully clarify that we strictly followed the anonymization policy**. The GitHub link we provided is anonymized and does not reveal any author identity or personal information.
>
> ## On Extending the Method to Other Video Editing Tasks
> We are the first to apply this type of bridge-based formulation to video object removal. In future work, we plan to extend this framework to other video editing tasks as well.

---

> > ### Author Rebuttal · Reviewer_MGpV · 2026-04-03
> >
> > The authors have addressed most of my concerns.

---

> > > ### Author Response · Authors · 2026-04-03
> > >
> > > We sincerely thank the reviewer for reviewing our rebuttal and acknowledging that most of your concerns have been addressed. We hope our clarifications have resolved the issues you raised. If so, we would be grateful if you would consider increasing the score.

---

### Official Review · Reviewer_xY1E · 2026-03-11

**Soundness:** 4
**Presentation:** 3
**Significance:** 3
**Originality:** 3
**Overall Recommendation:** 5
**Confidence:** 4

**Summary:**

This paper proposes BridgeRemoval, a stochastic-bridge framework for video object removal that reformulates the task as video-to-video translation instead of standard noise-to-data diffusion.  The author claims that the source video itself should be treated as a strong structural prior, so that the model can better preserve non-masked regions while removing the target object. They introduce a bridge-based formulation for video object removal and show that this design can produce strong quantitative performance together with very impressive visual quality. I am positive on the paper, mainly because the qualitative results are genuinely strong and the bridge formulation is a reasonable and effective direction for this task, although the use of Adaptive Mask Modulation is not yet fully motivated.

**Compliance With Llm Reviewing Policy:**

Affirmed.

**Final Justification:**

The rebuttal addressed all of my concerns, and I will maintain my Accept rating.

**Key Questions For Authors:**

Can the authors better justify why AMM is the appropriate way to address the rigidity introduced by the source prior? In particular, why should spatial affine modulation of patch embeddings be preferred over simpler mask-aware conditioning or gating mechanisms?


How much of the final gain comes from the bridge formulation alone? A clearer decomposition between the benefit of the stochastic bridge and the benefit of AMM is needed.

**Limitations:**

yes

**Strengths And Weaknesses:**

**Strengths**

The visual results are excellent and very convincing. Across the qualitative comparisons and additional examples, the method removes objects cleanly while keeping the background coherent and temporally stable, and the user study also shows a clear preference for the proposed method over prior baselines.


The main technical idea is clean and well aligned with the task. Using a stochastic bridge from source video to target video is a natural alternative to starting from Gaussian noise, and the quantitative results suggest that this design gives a good balance between object removal quality and background preservation.

**Weaknesses**

The motivation for AMM is not fully developed. The paper says that a strong bridge prior can make large-object removal harder, and AMM is introduced to relax this rigidity, but the argument remains fairly intuitive and does not clearly explain why this specific scale-and-shift modulation is the right mechanism.

---

> ### Author Rebuttal · Authors · 2026-03-29
>
> We sincerely appreciate your recognition of the strengths and effectiveness of our work and the valuable suggestions to help us improve that. We are happy to have a discussion and hope it could address your concerns.
>
> ## On the Motivation and Novelty of AMM
>
> We agree that the motivation of AMM should be clarified more explicitly. A key point is that AMM is not introduced as a replacement for mask-aware conditioning. Our model already uses mask-aware conditioning by concatenating the mask latent with the source latent as input condition. However, we found that this input-level conditioning alone is often insufficient when removing large objects: although it tells the model where editing should happen, the strong bridge prior can still overly preserve source features inside the masked region. This is exactly why we introduce AMM on top of mask-aware conditioning.
>
> The role of AMM is therefore to modify the feature computation itself, rather than only provide mask information at the input. Concretely, AMM applies spatially-varying affine modulation $h' = h \odot (1+\gamma) + \beta$, which is closely related to FiLM-style feature-wise linear modulation [1]. This form is appropriate here because the bridge prior creates a spatially non-uniform requirement: outside the mask, source fidelity should remain strong, while inside the mask, source-dominant features should be relaxed to enable complete removal and content regeneration. The multiplicative term $(1+\gamma)$ locally attenuates or preserves existing features, while the additive term $\beta$ injects mask-dependent corrective signals.
>
> This design is also well matched to the stochastic bridge formulation. Because the bridge starts from the source video as the prior state, it naturally encourages source-faithful reconstruction throughout the trajectory. This is desirable in unmasked regions, where preserving the original background is important, but it can be overly restrictive inside the masked region, especially for large-object removal. AMM addresses this spatial mismatch at the feature level: outside the mask, it can remain close to identity so that source fidelity is preserved; inside the mask, it can attenuate source-dominant features through $\gamma$ and inject region-specific corrective signals through $\beta$, thereby making content replacement easier.
>
> For this reason, we believe AMM is an appropriate mechanism in our setting. It does not replace mask-aware conditioning; rather, it complements it by turning mask information into explicit, spatially-varying feature modulation throughout the network.
>
> ## Ablation on the Contributions of the Bridge and AMM
>
> To more clearly decompose the gain from the stochastic bridge formulation and the gain from AMM, we added additional ablation experiments under a fully aligned training setup. Specifically, we trained a Wan2.1-1.3B baseline without the bridge strategy, using the same training set and the same model capacity for a fair comparison.
> From the table, adding the SDE bridge on top of the Wan2.1-1.3B baseline already leads to clear improvements in unmasked region preservation, with PSNR increasing from 30.7283 to 32.9215, MSE decreasing from 68.2547 to 48.6971, and LPIPS decreasing from 0.0453 to 0.0439. After further adding AMM, the performance improves again, especially on CLIP-T, which increases from 0.2846 to 0.2922. This indicates that AMM is particularly helpful for removing large objects while maintaining semantic consistency. Qualitative comparisons are provided in Figure 4 of the appendix, and we provide more qualitative results in https://bridgeremoval.github.io/rebuttal/ (See **Figure11,12,13**).
>
> Please feel free to tell us if you still have any concern and we are willing to have a further discussion.
>
> | Method | CLIP-T ↑ | CLIP-F ↑ | PSNR ↑ | MSE ↓ | LPIPS ↓ |
> |---|---:|---:|---:|---:|---:|
> | Wan2.1-1.3B | 0.2810 | 0.9815 | 30.7283 | 68.2547 | 0.0453 |
> | Wan2.1-1.3B+SDE Bridge | 0.2846 | 0.9880 | 32.9215 | 48.6971 | 0.0439 |
> | Wan2.1-1.3B+SDE Bridge+AMM (BridgeRemoval) | **0.2922** | **0.9917** | **33.2394** | **46.3525** | **0.0415**|
>
> Reference
>
> [1] Perez, Ethan, et al. "FiLM: Visual reasoning with a general conditioning layer." Proceedings of the AAAI conference on artificial intelligence. 2018.

---

> > ### Author Rebuttal · Reviewer_xY1E · 2026-04-03
> >
> > Thank you for the response. My problems have been resolved.

---

> > > ### Author Response · Authors · 2026-04-03
> > >
> > > We sincerely thank the Reviewer for the positive feedback and for acknowledging that all concerns have been adequately addressed. We truly appreciate your constructive comments and the time spent reviewing our work.

---

### Decision · Program_Chairs · 2026-04-30

**Decision:**

Accept (regular)

**Comment:**

The final reviews were one Accept, two WA's and one WR. However, reviewer Rxds with the WR rating had indicated in their final rebuttal response that the authors had addressed most of their concerns and preferred to accept the paper --- in this instance, the AC would go with the reviewer's statement and assumed an intention to raise the score to WA. Given this interpretation, the reviewers have unanimously sided with accepting the paper. The reviewers generally found the paper to be well-written, provides for a new benchmark for video object removal, proposes a well-designed approach that is useful, and has strong results. There was some initial concerns about the level of novelty (e.g. vs DDBM) and underlying motivation, but the reviewers considered these to be resolved by the rebuttal. The AC agrees and recommends an accept.